# Uncoupling evolutionary changes in DNA sequence, transcription factor occupancy and enhancer activity

**Pierre Khoueiry[1][†], Charles Girardot[1], Lucia Ciglar[1], Pei-Chen Peng[2], E Hilary Gustafson[1], Saurabh Sinha[1,2], Eileen EM Furlong[1]\***

[1]European Molecular Biology Laboratory, Genome Biology Unit, Heidelberg, Germany; [2]Carl R. Woese Institute of Genomic Biology, University of Illinois, Champaign, United States

**Abstract** Sequence variation within enhancers plays a major role in both evolution and disease, yet its functional impact on transcription factor (TF) occupancy and enhancer activity remains poorly understood. Here, we assayed the binding of five essential TFs over multiple stages of embryogenesis in two distant *Drosophila* species (with 1.4 substitutions per neutral site), identifying thousands of orthologous enhancers with conserved or diverged combinatorial occupancy. We used these binding signatures to dissect two properties of developmental enhancers: (1) potential TF cooperativity, using signatures of co-associations and co-divergence in TF occupancy. This revealed conserved combinatorial binding despite sequence divergence, suggesting protein-protein interactions sustain conserved collective occupancy. (2) Enhancer in-vivo activity, revealing orthologous enhancers with conserved activity despite divergence in TF occupancy. Taken together, we identify enhancers with diverged motifs yet conserved occupancy and others with diverged occupancy yet conserved activity, emphasising the need to functionally measure the effect of divergence on enhancer activity.

**\*For correspondence:** furlong@embl.de

**Present address:** [†]Department of Biochemistry and Molecular Genetics, American University of Beirut, Beirut, Lebanon

**Competing interests:** The authors declare that no competing interests exist.

## Introduction

Transcription factors (TFs) act largely through enhancer elements - modular sequences that instruct genes when and where to be expressed (*Levine and Davidson, 2005*; *Slattery et al., 2014*; *Spitz and Furlong, 2012*) and provide robustness and precision to developmental programs (*Cannavò et al., 2016*; *Frankel et al., 2010*; *Perry et al., 2010*). Changes in TF binding sites within an enhancer can lead to phenotypic differences between individuals (*Prud'homme et al., 2007*; *Rogers et al., 2013*), and between species (*Levine and Davidson, 2005*; *Wray, 2007*). In sticklebacks, for example, the recurrent deletion of an entire enhancer driving pelvic *pitx* expression leads to the lack of spines in several freshwater populations (*Chan et al., 2010*; *Shapiro et al., 2004*). In other cases, however, changes in enhancer sequence appear to have limited impact on phenotype. The loss or gain of TF binding sites in the *otx* enhancer in ascidians (*Oda-Ishii et al., 2005*), the *Endo 16* enhancer in sea urchins (*Balhoff and Wray, 2005*), and the *eve* enhancer in *Drosophila* (*Ludwig et al., 1998*), for example, have only marginal effects on enhancer activity. This often-surprising gap between changes in genotype and phenotype in developmental contexts highlights the current challenge in understanding both enhancer function and evolutionary constraints.

Several models have been proposed to explain enhancer function. The enhanceosome describes elements that are bound by many TFs through cooperative interactions facilitated by the very precise relative orientation, spacing and helical phasing of the TF's binding sites (TFBSs) within the enhancer. Mutations or sequence variants that affect either the motif quality or the motif

arrangement results in loss of enhancer activity. The best studied, and perhaps only true enhancesome, is the interferon beta enhancer, a 57 bp core element with a very strict arrangement of TFBS (*Panne et al., 2007*; *Thanos and Maniatis, 1995*). At the other extreme the 'billboard' model (*Arnosti and Kulkarni, 2005*; *Kulkarni and Arnosti, 2003*) proposes that TFs bind additively or in subgroups to an enhancer with relatively independent effects on gene expression, and therefore have flexibility in the relative positions of their TFBSs (*Khoueiry et al., 2010*). We recently proposed the TF collective model, where TFs bind to enhancers cooperatively through a combination of protein::DNA and protein::protein interactions allowing for a flexible combination of TFBSs. Unlike billboard enhancers, a TF collective does not require the presence of motifs for every TF to be occupied by the full collective, as observed in cardioblast (*Junion et al., 2012*), leg precursor (*Uhl et al., 2016*) and dopaminergic neuron (*Doitsidou et al., 2013*) enhancers. One prediction from this model is that such enhancers should tolerate some evolutionary changes in TFBS content, while having little or no effect on TF occupancy across species, as recently observed for the *distalless* enhancer occupied by a Hox TF collective (*Uhl et al., 2016*). If the TF collective model holds true more globally, it provides one explanation for why it is currently not possible to accurately predict which genetic variants will affect enhancer function based on DNA sequence alone.

As a step towards linking sequence variation to enhancer function, several studies have coupled sequence analyses with in vivo measurements of TF occupancy between species, in either developing embryos (*Bradley et al., 2010*; *He et al., 2011*; *Ni et al., 2012*; *Paris et al., 2013*) or differentiated tissues (*Ballester et al., 2014*; *Odom et al., 2007*; *Schmidt et al., 2010*; *Wilson et al., 2008*). The results revealed marked differences in the extent to which TF binding is conserved, highlighting a complex interplay between tissue type, cell state (unspecified versus differentiated), the TF analyzed and evolutionary distance, as well as potential technical issues such as antibody specificity across species and the difficulty to assign orthologous non-coding regions (reviewed by [*Sakabe and Nobrega, 2013*; *Villar et al., 2014*]). For example,~60% of sites bound by the developmental master regulator Twist (Twi) have conserved occupancy between *Drosophila melanogaster* and *pseudoobscura* (*He et al., 2011*), an evolutionary distance comparable to that between human and chicken (estimated 1.1 substitutions per neutral site (*Stark et al., 2007*)). However, the level of conserved binding dropped significantly when considering more distant species, *D. melanogaster* and *D. virilis* (1.4 substitutions per neutral site), for factors involved in anterior-posterior patterning in the early *Drosophila* embryo (*Paris et al., 2013*). In vertebrates, extensive divergence in TF occupancy was observed in differentiated tissues. For instance, in the adult liver, only 14% of CEBPA binding are conserved between human and mouse (*Schmidt et al., 2010*), species separated by ~60 million years (0.5 substitutions per neutral sites [*Stark et al., 2007*]). The conservation for CEBPA binding drops to ~2% when the authors compared between very distant species, human and chicken, separated by ~300 million years. In contrast, the binding of CTCF is highly conserved (*Schmidt et al., 2012*). In each study the authors used different methods to define conserved TF binding. When the datasets were re-analyzed using a unifying statistical framework, which alleviates differences in analytical methods (*Carvunis et al., 2015*), the rate of evolution or divergence in TF binding are similar between vertebrates and flies, suggesting that transcriptional networks evolve at a common rate. While these studies have greatly improved our understanding of the rate at which TF occupancy events evolve, the functional consequences of these changes to enhancer activity, and therefore developmental programs, remain poorly understood.

Here, we used cross-species TF occupancy to assess inherent properties of enhancers, including testing the evolutionary flexibility predicted by the TF collective model. To initiate the study, we performed ChIP-seq on five essential developmental TFs across multiple stages of embryonic development in *D. virilis* and compared their binding signatures to orthologous stages in *D. melanogaster*. These two species are highly diverged (neutral substitutions per base of 1.4) with an estimated molecular distance ~3 times that between human and mouse (*Stark et al., 2007*).The five factors examined, Twi, Mef2, Tinman (Tin), Bagpipe (Bap) and Biniou (Bin), are the major drivers of the subdivision of the mesoderm into different muscle primordia and form part of a highly interconnected gene regulatory network (*Figure 1a*) (*Bonn and Furlong, 2008*; *Jakobsen et al., 2007*; *Liu et al., 2009*; *Sandmann et al., 2007, 2006*; *Zinzen et al., 2009*). This thereby provides an ideal system to investigate divergence and conservation along different layers of a developmental network (*Figure 1a,b*). Using these signatures of TF occupancy, we identified a conservative set of ~2800 orthologous *cis*-regulatory elements and examined their combinatorial and dynamic occupancy

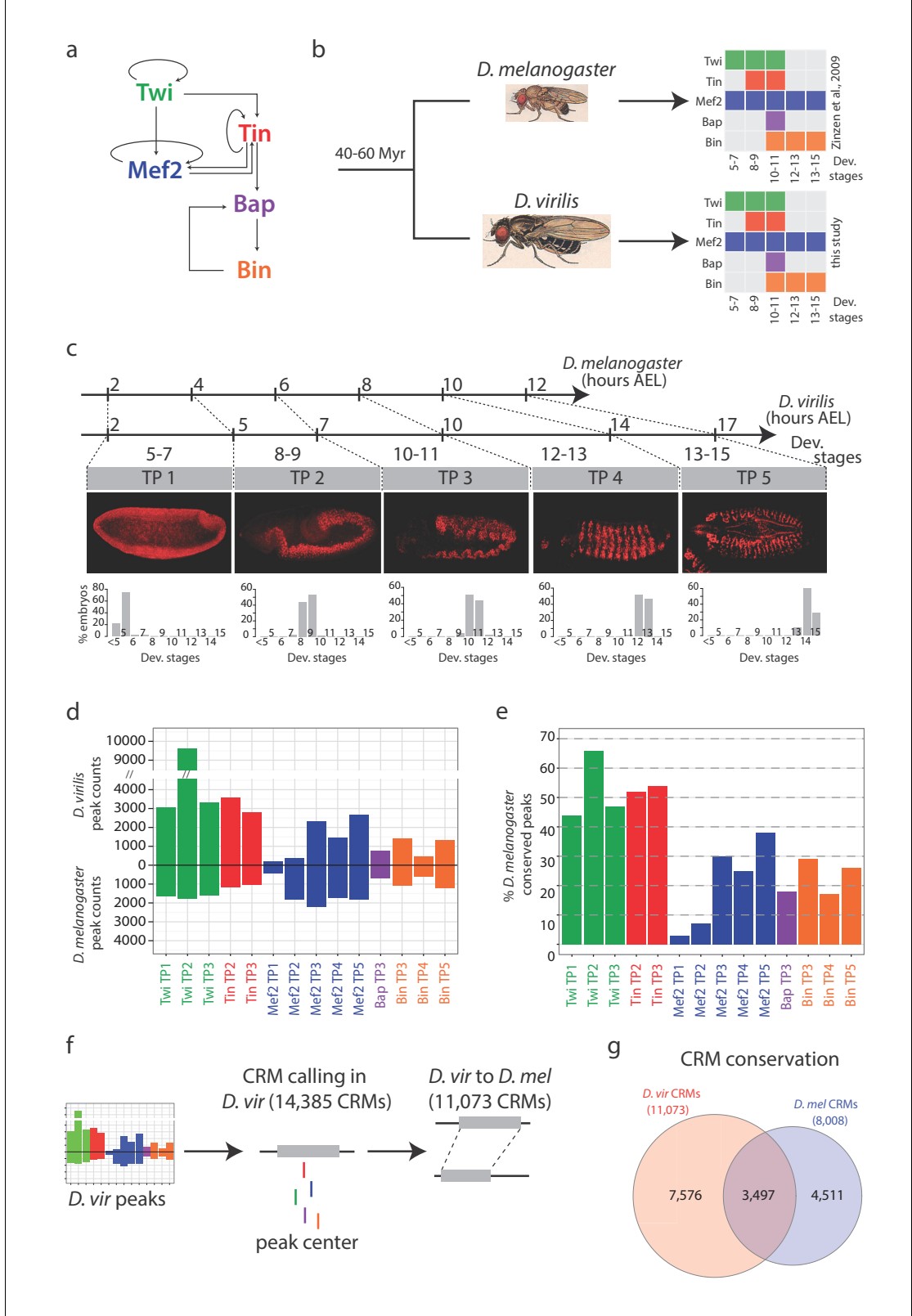

**Figure 1.** Conservation in TF occupancy across developmental and evolutionary time-scales. (**a**) The regulatory network linking 5 mesodermal transcription factors essential for mesoderm specification. (**b**) *D. melanogaster* and *D. virilis* ChIP were performed using species-specific antibodies for Twi, Tin, Bap and Bin. Colored boxes illustrate TFs and time points analyzed, with developmental stages on the x-axis. (**c**) The scale illustrates equivalent time

*Figure 1 continued on next page*

*Figure 1 continued*

points for corresponding stages between *D. melanogaster* and *D. virilis*. Axis labels correspond to hours of development, after egg laying (AEL) and numbers below axis correspond to developmental stages and Time Points (TP 1–5). Confocal images of embryos with Mef2 immuno-stains at the most representative stage for each time-point, orientated anterior-left, dorsal-up. Barplots show percentage of embryos at each stage (x-axis) for each TP, averaged over two replicates. (**d**) Number of significant ChIP peaks in each condition in both species. TP1, TP2, etc, correspond to Time point 1, Time point 2 etc. *D. melanogaster* peaks are from (*Zinzen et al., 2009*). (**e**) Percentage of conserved peaks between *D. melanogaster* and *D. virilis*. (**f**) Peak summits in close proximity were used to define *cis*-regulatory modules (CRMs, see Materials and methods). (**g**) Venn diagram of orthologous CRMs conserved between both species; 3,497 CRMs (*D. mel* CRMs overlapping more than one *D. vir* CRMs are counted once).

The following figure supplements are available for figure 1:

**Figure supplement 1.** TF expression pattern in *D. virilis* embryos.

**Figure supplement 2.** PWMs of orthologous transcription factors are highly similar.

across five embryonic time-points. We first examined how the loss of a given TF binding event affects the binding of other TFs that co-occupy the same element using evolutionary divergence as a tool to identify potential cooperative TF interactions. We found inter-species divergence in TF binding associated with either the loss of that TF's motif or the loss of binding of other TFs, as also observed in studies in closely related *Drosophila* (*Bradley et al., 2010*; *He et al., 2011*) and mouse (*Odom et al., 2007*; *Schmidt et al., 2010*; *Stefflova et al., 2013*) species. In addition, we also observed the converse, conserved TF binding associated with reduced sequence dependency, when the TF is combinatorially bound with other factors. This is consistent with the inherent flexibility predicted by the TF collective model, and provides the first large-scale insights into the extent to which this can occur.

Second, we directly assessed the impact of evolutionary changes in TF occupancy on enhancer function by examining spatio-temporal activity of orthologous enhancer pairs. This cross-species in-vivo enhancer testing revealed conserved activity despite extensive divergence in TF occupancy. Taken together, our results indicate many instances where evolutionary changes in enhancer motifs, or in enhancer TF occupancy, have little effect on enhancer activity, highlighting the inherent difficulty in predicting the functional effects of *cis*-regulatory variants on enhancer activity and therefore gene expression.

## Results

### Interspecies ChIP for five developmental TFs across five stages of embryogenesis

To accurately compare cross-species signatures of TF binding during embryogenesis, it is essential to compare equivalent developmental stages. The life cycle from egg lay to adult between *D. virilis* and *D. melanogaster* is temporally different, with embryogenesis taking ~10 hr longer in *D. virilis* compared to *D. melanogaster* (*Markow et al., 2009*). We therefore first determined equivalent embryo collection windows by comparing developmental stages of timed collections between both species using the expression of Mef2 protein (*Figure 1c*). In *D. melanogaster*, Mef2 is very dynamically expressed throughout the mesoderm during all stages of its specification and subsequent differentiation and thereby serves as a very informative molecular indicator of developmental stage. By counting the distribution of stages between multiple, overlapping collections, we identified the optimal time intervals to obtain an equivalent distributions of embryonic stages across the five developmental time windows of interest (*Figure 1c*). These time-intervals were recently confirmed in an independent study (*Kuntz and Eisen, 2014*).

To circumvent potential technical inconsistencies due to reduced antibody affinity across species, we generated new antibodies for four TFs directed against the full-length *D. virilis* proteins: Tin, Twi, Bin and Bap. For these TFs, ChIP was therefore performed with species-specific antibodies. For

Mef2, the *D. melanogaster* antibody performed equally well using *D. virilis* or *D. melanogaster* embryos for both ChIP-qPCR (Materials and methods) and immunostaining (*Figure 1—figure supplement 1*). Importantly, all five antibodies gave the expected, highly specific and dynamic patterns of expression in *D. virilis* embryos (*Figure 1—figure supplement 1*), demonstrating the specificity of the antibodies and the conservation in the spatio-temporal expression patterns of these TFs.

ChIP-seq was performed in *D. virilis* embryos across five time windows of embryogenesis, equivalent to those we previously used to analyze these transcription factors occupancy in *D. melanogaster* (*Zinzen et al., 2009*) (*Figure 1b,c*). The number of time-points examined for each TF reflects the developmental stages that the TF is expressed, totaling 14 conditions (TF and time point). For all TFs across all corresponding time-points, two independent biological replicates were generated, using two independent sera when possible (Twi, Mef2), leading to a total of 28 *D. virilis* ChIP-seq datasets and 10 stage-matched genomic input controls (*Figure 1b*). Replicates were highly reproducible for both raw reads (median Pearson correlation Rho = 0.83) and significant TF peaks (median Pearson correlation Rho = 0.82). In addition, de novo motif discovery systematically identified the expected motifs enriched under the TF peak summits, attesting to the quality of the *D. virilis* ChIP-data (*Figure 1—figure supplement 2*). The number of high confidence peaks, called at a 1% Irreproducible Discovery Rate (IDR, a measure ensuring equivalent reproducibility between replicas) threshold, corresponding to peak calling q-value <0.001 in all conditions (Materials and methods). The number of peaks was generally higher for each TF in the *D. virilis* ChIP-seq compared to their corresponding *D. melanogaster* ChIP-chip data (*Zinzen et al., 2009*) (*Figure 1d*, *Supplementary file 1*), potentially reflecting the larger size of the *D. virilis* non-coding genome (which we estimate is approximately 50% larger) or the higher sensitivity of ChIP-seq versus ChIP-chip. Importantly, the general trend in the relative number of peaks per TF across time-points appears largely conserved between both species (*Figure 1d*).

## Conservation of TF binding is factor-dependent

Evolutionary changes in non-coding DNA, including TF binding site turnover and high rates of insertions and deletions can complicate the identification of orthologous TF binding events (*Kalay and Wittkopp, 2010*; *Moses et al., 2006*). Given the evolutionary distance between *D. melanogaster* and *D. virilis*, we determined ChIP peaks in each species separately, defining species-specific significantly bound regions. The peaks identified in *D. virilis* were then translated onto *D. melanogaster* coordinates using pslMap (*Zhu et al., 2007*), resulting in 89% of bound peaks translated (Materials and methods, *Supplementary file 1*).

To evaluate the conservation and divergence of TF occupancy, we first examined cross-species TF occupancy in the 14 TF-conditions separately (*Figure 1b*), by calculating the fraction of *D. melanogaster* peaks that overlap with *D. virilis* peaks for a specific TF at the same developmental time-point. This revealed no obvious relationship between binding conservation and the underlying biological function of the different TFs, with the interesting exception of Mef2. Early in embryogenesis Mef2 binds to a small number of regions and the binding is highly diverged amongst species. This could represent a shift (or delay) in the initiation of Mef2 binding between species (*Liotta et al., 2007*), rather than in Mef2 expression (which is conserved as seen in *Figure 1c* and *Figure 1—figure supplement 1*). In contrast, by mid- and late- embryogenesis both the number and conservation in Mef2 binding is substantially higher (*Figure 1d,e*; compare TP1, 2 to TP3-5), matching the essential role of Mef2 in muscle differentiation at these later embryonic stages. In contrast, the occupancy of Twi and Tin are highly conserved over all time-windows examined, with on average ~50% conservation in binding. These results for Twi are in line with previous comparisons between *D. melanogaster* and *D. pseudoobscura* (*He et al., 2011*). Both Twi and Tin are essential for cell fate specification of the trunk and dorsal mesoderm, respectively, and both factors regulate extensive downstream transcriptional cascades (*Bonn and Furlong, 2008*; *Liu et al., 2009*; *Sandmann et al., 2007*). In contrast, only ~19% of Bap's and 26% of Bin's occupancy is conserved (*Figure 1e*, *Supplementary file 1*), yet both TFs are essential for the specification of the visceral muscle primordial, and Bin for its subsequent differentiation (*Jakobsen et al., 2007*; *Zaffran et al., 2001*); loss-of-function *bap* or *bin* mutant embryos develop no gut muscle and are therefore embryonic lethal (*Azpiazu and Frasch, 1993*; *Zaffran et al., 2001*). It is interesting to note that the orthologs of Bin, FoxF TFs, are also essential for gut muscle development in both mice (*Ormestad et al., 2006*) and Xenopus

(*Tseng et al., 2004*). Therefore, although these TFs have an ancient essential role in visceral muscle development, their enhancer occupancy appears to be rapidly evolving.

## Conserved enhancer occupancy is associated with enhancer function

TFs rarely act alone, but rather bind to developmental enhancers in a highly combinatorial, and often cooperative manner (*Spitz and Furlong, 2012*). Evolutionary constraints, therefore, likely act at the level of enhancers, selecting for global enhancer composition rather than on the positioning of individual binding sites (*Lusk and Eisen, 2010*). The five TFs assessed here are known to combinatorially regulate enhancers during mesoderm and muscle development (*Jakobsen et al., 2007*; *Liu et al., 2009*; *Sandmann et al., 2007*, *2006*; *Zinzen et al., 2009*). We therefore next examined evolutionary changes in TF occupancy at the level of combinatorial occupancy, rather than at individual bound peaks.

We previously defined *cis*-regulatory modules (CRMs) targeted by these five TFs in *D. melanogaster* by clustering ChIP peak summits within 100 bp of each other, collapsing the 19,522 significant ChIP-peaks into 8008 ChIP defined putative CRMs (ChIP-CRMs, [*Zinzen et al., 2009*]), henceforth called CRMs. Importantly, 95% of the elements tested (with an average size of 271 bases), function as developmental enhancers in vivo in stable transgenic embryos, as seen from a number of studies (*Cannavò et al., 2016*; *Ciglar et al., 2014*; *Junion et al., 2012*; *Zinzen et al., 2009*). Here, we grouped the set of 33,285 significant *D. virilis* ChIP peaks, spanning all TFs and all time-points, into a high-confidence set of 14,385 non-overlapping CRMs, with an average size of 273 bases (Materials and methods, *Supplementary file 2*); 11,073 (77%) of which could be translated to *D. melanogaster* coordinates using pslMap (*Figure 1f*, *Supplementary file 3*, Materials and methods).

Fewer than half of *D. melanogaster* CRMs (44%; 3497/8008, *Figure 1g*), overlap a *D. virilis* CRM (median overlap of 170 bp, corresponding to 62% median CRM length): 2846 CRMs are in the orthologous region in a one-to-one relationship (*Figure 1f*), while an additional 651 *D. melanogaster* CRMs overlap two or more adjacent *D. virilis* CRMs. Interestingly, this overlap is very similar to that observed in humans, where less than half of CRMs were found in an orthologous position in another vertebrate species (*Ballester et al., 2014*). Although additional non-overlapping orthologous enhancer pairs surely exist, we focused all subsequent analysis on this conservative set of 2846 orthologous elements. Given the large evolutionary distance between the species, we reasoned this should avoid inflating estimates of divergence in TF occupancy (enhancer input) due to mis-matched non-orthologous enhancer pairs.

Of the 2846 orthologous CRMs, only 14% (404/2846 elements) are bound by a single TF at a single time-point in both species, with the vast majority (86%, 2442 elements) being bound by multiple TFs and/or at multiple developmental stages in one or the other species. To quantify the overall conservation in combinatorial binding of each orthologous CRM pair at each time-point, we use Jaccard similarity coefficient (J), which ranges from 0 (none of the significant TF binding events are shared) to 1 (all TF binding across all time-points (14 conditions) are shared) (*Figure 2a*). Based on this systematic indexing, the 2442 cross-species multi-ChIP peak CRM pairs were distributed into three binding conservation categories; those with strong (J >= 0.5; 812 CRMs, *Supplementary file 4*), intermediate (0 < J < 0.5; 1130 CRMs, *Supplementary file 5*) and no binding conservation (J = 0; 500 CRMs, *Supplementary file 6*) (*Figure 2a*). Strong and intermediate categories thereby reflect CRMs with greater or fewer than 50% of binding events conservation, respectively.

Enhancers with conserved TF binding (both strong and intermediate levels) have a higher number of total TF binding events in *D. melanogaster*, i.e. more combinatorial occupancy compared to ones with no conservation (*Figure 2b*, Chi-square test p-value=$5e^{-04}$ for each of the three pairwise comparisons), as expected (*Paris et al., 2013*). *D. mel* CRM-3805, for example, is bound by all five factors at multiple stages and is therefore occupied in all 14 developmental conditions examined in *D. melanogaster* (*Figure 2c*). This extensive occupancy is highly conserved in *D. virilis* (*D. vir* CRM-11767) for all five factors across almost all time-points examined, with the exception of Bap (*Figure 2c*). In contrast, *D. mel* CRM-6503, is bound only by Mef2 at four stages of embryogenesis in *D. melanogaster*, but has no significant Mef2 binding in *D. virilis* (*D. vir* CRM-14210) (*Figure 2c*). Interestingly, it is occupied by Twi at the three early stages in *D. virilis*, but not in *D. melanogaster*, suggesting either a change in the activity of the enhancer itself or a change in the functional TF required to achieve the same enhancer activity (e.g. a TF with similar expression).

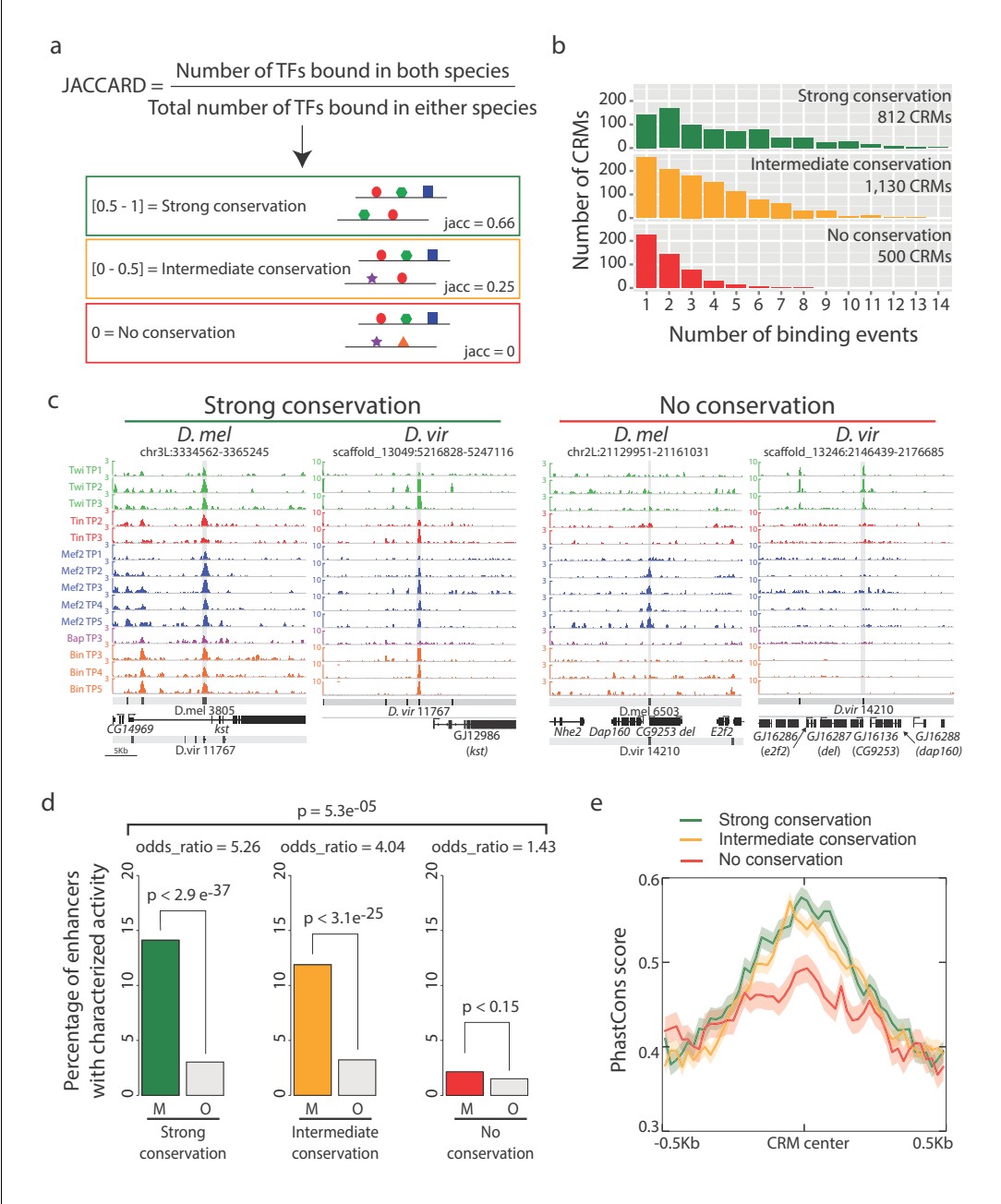

**Figure 2.** Enhancers with conserved binding are associated with functional elements. (**a**) Jaccard similarity index was used to categorize CRMs into three groups based on their TF binding conservation. The schematic represents CRM pairs for each of the three groups with hypothetical combinatorial binding and the associated Jaccard Index. (**b**) Barplot representing the complexity of *D. melanogaster* CRMs, in term of number of binding events (x-axis), for each CRM binding conservation category. (**c**) Genome browser visualization of strongly conserved (left) and non-conserved (right) CRM pairs. *D. virilis* (*D. vir*) tracks display RPM normalized, input subtracted ChIP-seq signal. *D. melanogaster (D. mel)* tracks show log2(IP/mock) ChIP-chip signal. In *D. mel*, both *D. mel* and translated *D. vir* CRMs are shown along with *D. mel* gene models. In *D. vir*, *D. vir* CRMs and genes are shown with *D. mel* orthologous gene name depicted between parenthesis. The same genomic window size was used for both species, indicated in the lower left corner. (**d**) Fraction of mesodermal ('M', coloured) and non-mesodermal ('O' for 'Other tissues', gray) enhancers overlapping CRMs from the three categories defined in (**a**). Overall p-value corresponds to a two-sided proportions test, p-values within categories correspond to Fisher exact test. (**e**) CRMs with strong or intermediate conservation in their TF binding events have stronger sequence constrains than those with no conserved binding, based on their average PhastCons conservation scores. Solid lines represent the mean and shading delineates standard error.

To assess the likely functional consequences of diverged versus conserved TF binding, we took advantage of a large collection of developmental enhancers with characterized spatio-temporal activity in vivo, using transgenic *D. melanogaster* embryos (*Bonn et al., 2012*; *Gallo et al., 2011*; *Kvon et al., 2014*). Enhancers were divided into two categories, those active in the same cell type as the TFs (mesoderm/muscle: 1254 enhancers) and enhancers active in any other tissue, excluding the mesoderm (2970 enhancers, Materials and methods, *Supplementary file 7*). Overlapping our inter-species CRMs bound by mesoderm/muscle TFs with these enhancers indicates that 494 (20.3%; 494/2442) CRMs in an orthologous pair have been experimentally tested in *D. melanogaster* and shown to function as developmental enhancers in vivo. Importantly, CRMs with strong and intermediate conservation in their occupancy (based on their Jaccard similarity coefficient) have significantly higher overlap with characterized mesodermal and muscle enhancers compared to CRMs with non conserved binding (*Figure 2d*, two-sided proportions test p-value=5.3e$^{-05}$). Conserved enhancer binding is therefore globally associated with enhancer function. It is interesting to note that CRMs with no conserved binding (J = 0) are still enriched for mesoderm/muscle activity, although to a much lesser extent than the two conserved categories (odds ratio of 1.43 vs 5.26 and 4.04 for the strong and intermediate conservation classes, respectively, *Figure 2d*). In addition, CRMs with strong and intermediate conserved binding show stronger sequence conservation (*Figure 2e*), suggesting that much of the conservation and divergence in occupancy is directed by changes in the DNA sequence within the regulatory element itself, which we examine in more detail below.

## Developmental enhancers preserve dynamic combinatorial patterns of TF occupancy over a large evolutionary time-span

We previously showed that TFs occupy enhancers in much more dynamic temporal windows than their expression patterns suggest (*Wilczyński and Furlong, 2010*). Twi, Tin, Mef2 and Bin, for example, each occupy their enhancers during *D. melanogaster* development in three temporal classes (*Jakobsen et al., 2007*; *Liu et al., 2009*; *Sandmann et al., 2007*, *2006*): continuously at all stages of development where they are expressed, and two interesting transient classes occupied only at early or late stages of development, despite the TFs being expressed at broader developmental windows. The occupancy of the two latter classes cannot be readily explained by the sequence motifs of the TFs themselves (*Wilczyński and Furlong, 2010*) and suggests cooperative binding with other TFs, which is supported by motif enrichment in each temporal class (*Wilczyński and Furlong, 2010*; *Yáñez-Cuna et al., 2012*). Taking advantage of the 2846 cross-species orthologous CRMs identified in this study, we can now assess if these dynamic patterns of TF occupancy are conserved, and explore how they might be regulated.

To more directly compare dynamic patterns of TF occupancy across this evolutionary time-scale, we used hierarchical clustering based on the quantitative ChIP signal from both species (Materials and methods). This minimizes thresholding issues and better reflects the dynamic range of TF occupancy as a quantitative continuum (*Biggin, 2011*). Although the *D. melanogaster* data is ChIP-chip and *D. virilis* ChIP-seq, the TF occupancy patterns from the chromatin immunoprecipitation are highly correlated (*Figure 3—figure supplement 1*).

The level of conserved occupancy for these orthologous enhancers is striking; CRM pairs have similar coordinated changes of the same TF coming either 'on' or 'off' certain elements during developmental progression in species that span over 40 million years of evolution (*Figure 3a*). For instance, one group of enhancers is coordinately bound by Twi only at the first two developmental stages in both species (early-only CRMs), while another group is bound by Mef2 or Mef2 and Bin at late stages (late-only) or bound by all factors at almost all stages in both species (continuous) (*Figure 3a*). In *D. melanogaster*, these temporal patterns in TF occupancy are highly correlated with both the timing of enhancer activity (*Jakobsen et al., 2007*) and the expression and developmental function of the neighboring gene (*Wilczyński and Furlong, 2010*). We now show that these dynamic combinatorial patterns are also maintained across large evolutionary distances, suggesting that these temporal patterns of TF occupancy are functionally important.

To determine how these temporal patterns of TF occupancy are regulated, we used sequence analysis to search for motifs for additional potential partner TFs. For each species, we selected elements that are bound at only early or late stages, in either *D. virilis* or *D. melanogaster* (Materials and methods) and then performed motif analysis on each species separately. Previously we, and Yanez-Cuna *et al.*, found many motifs specifically enriched in *D. melanogaster* Twist early- versus

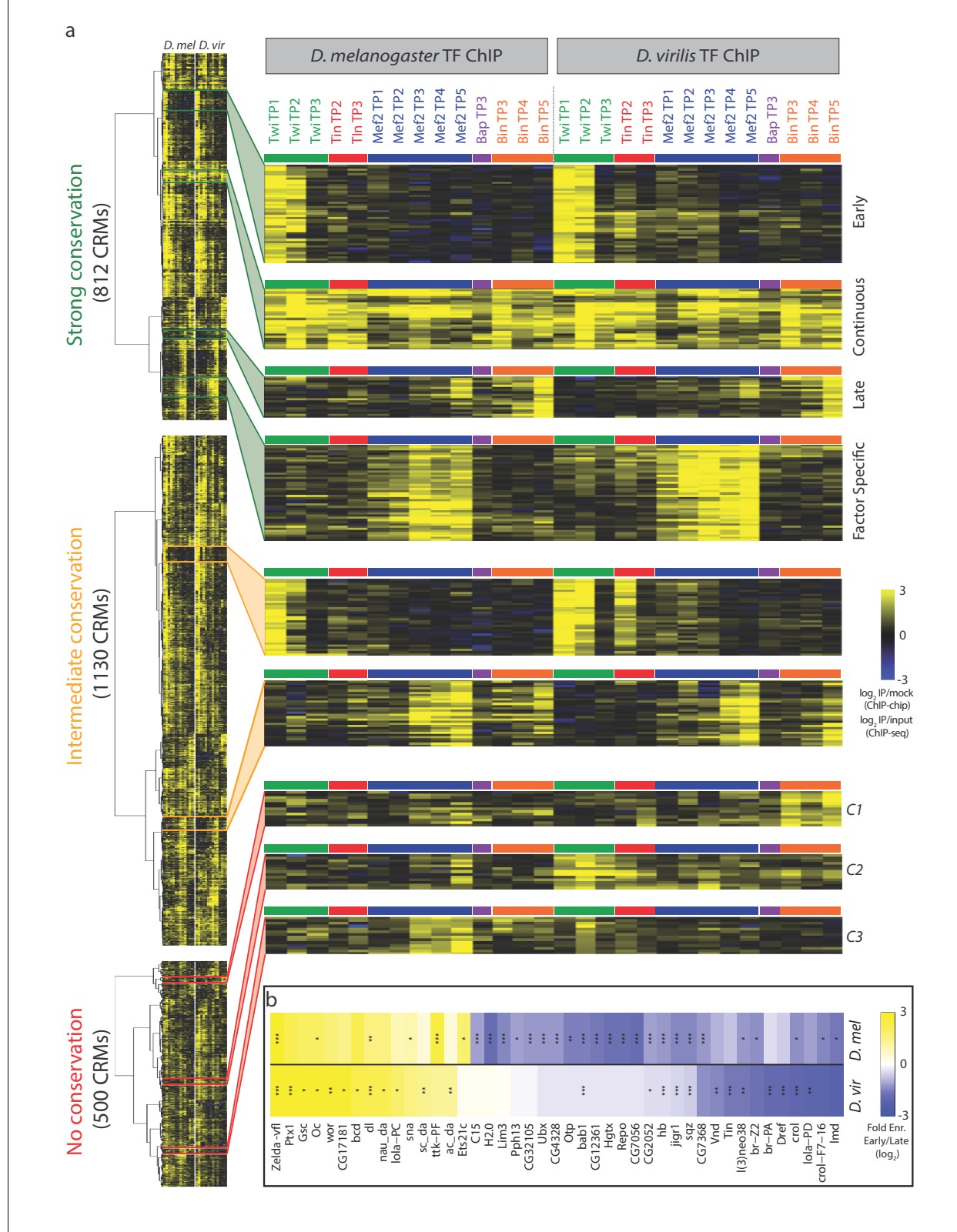

**Figure 3.** Dynamic patterns of developmental enhancer occupancy across distantly related species. (a) Hierarchical clustering of ChIP signal of CRMs with high, intermediate and low binding conservation, as classified using Jaccard distance (*Figure 2*). Each row corresponds to an orthologous CRM, and each column to a condition, with 14 *D. melanogaster* conditions followed by their equivalent 14 *D. virilis* conditions. The vertical dashed line delimits data from both species, while the horizontal colored bars indicate the TF. All three heatmaps are visualized using the quantitative ChIP-signal

*Figure 3 continued on next page*

*Figure 3 continued*

shown in the scale in lower right. (**b**) Differential motif analysis on early vs late Twi bound enhancers indicates that the regulation of these elements is largely conserved. Heatmap shows fold enrichment (early vs late bound enhancers, log2) of all motifs with >2 fold significant enrichment in one of the two species. *p<0.05; **p<0.01; ***p<0.001 indicates significance in that species. Color indicates motif enrichment, which is generally in the same direction in both species, even when significant in only one species.
The following figure supplement is available for figure 3:

**Figure supplement 1.** ChIP-chip and ChIP-seq signals are highly correlated.

late-bound enhancers (*Wilczyński and Furlong, 2010*; *Yáñez-Cuna et al., 2012*). Here, using a more expanded set of PWMs, we rediscover many of these motifs in the *D. melanogaster* elements and now determine if these motifs are evolutionarily conserved within the *D. virilis* Twist early- versus late-bound elements. Almost all motifs specifically enriched in the early-bound CRMs are present in both species (*Figure 3b*), suggesting a role, together with Twist, in the regulation of early-bound CRMs. This includes motifs for Zelda and Dorsal, two TFs functionally required for Twi binding to a number of early blastoderm enhancers (*Jiang and Levine, 1993*; *Yáñez-Cuna et al., 2012*), in addition to many other motifs that may indicate new partner TFs. Interestingly, the Twist-late bound enhancers have more divergence in the additional motifs enriched in each species (*Figure 3b*), which may reflect the cellular heterogeneity that is being specified during this developmental stage. This motif analysis on CRMs with evolutionarily conserved patterns of TF occupancy can therefore help discriminate between functional and likely non-functional motifs, and provides a useful tool to identify additional potential partner TFs that can form the basis of future studies. It also indicates that the underlying biology of how these two groups of enhancers are regulated goes beyond Twist, the factor that we have experimentally measured here.

The non-conserved set (J = 0) contains interesting orthologous enhancers where the identity of the significantly bound TF(s) has changed during evolution (*Figure 3a*). A set of enhancers bound by Bin in *D. virilis*, for example, is bound by Mef2 at these stages of development in *D. melanogaster* (*Figure 3a*, cluster C1). Another cluster of orthologous CRMs is bound exclusively by Mef2 in *D. melanogaster*, but by Twi and Tin in *D. virilis* (*Figure 3a*, cluster C2). A third set is bound by Mef2 in *D. melanogaster* and Twi in *D. virilis* (*Figure 3a*, cluster C3). These switches in TF occupancy may be the result of adaptive changes, as reported in yeast (*Hogues et al., 2008*) and suggests that either (1) the swapped TFs can perform the same function and thereby compensate for the TF loss resulting in a conserved enhancer activity or (2) this change in TF occupancy will lead to a dramatic change in enhancer function between these two species. Both scenarios are assessed below, and may be missed in studies examining the occupancy of a single factor in isolation.

## TF binding site divergence is tolerated in the context of collective TF binding

Loss of TF binding in inter- and intra-species analysis is often linked to sequence changes in the motif for the bound TFs (*He et al., 2011*; *Kasowski et al., 2010*; *Schmidt et al., 2010*; *Zheng et al., 2010*). To evaluate this for the occupancy of these five developmental factors, we compared overall motif density and conservation between CRMs with conserved versus non-conserved TF binding. As regulatory elements bound by more TFs tend to be more conserved (*Ballester et al., 2014*; *Hemberg and Kreiman, 2011*), we assessed motif properties across three classes of orthologous CRMs: (1) globally across all CRMs bound by four or five TFs in *D. melanogaster* (267 CRMs; called High Bound), (2) CRMs bound by two or three TFs in *D. melanogaster* (1004 CRMs; called Low Bound) and (3) CRMs with high-confidence binding by only one TF in *D. melanogaster* (1475 CRMs, called Singletons)(Materials and methods). For each category, CRMs where a TF binding was absent in *D. virilis* but present in *D. melanogaster (lost binding)* were compared to those where the binding was present in both species (*conserved binding*).

In all cases, the loss of TF binding is associated with a significantly lower density of that TF's motif (p-values < 1e-7, binomial test) for each of the five TFs in *D. virilis* (*Figure 4a*, compare circles to triangles). Notably, when TF binding is lost, the motif densities drop to background averages (*Figure 4a*, compare triangles to the bars), indicating that loss of TF occupancy is generally

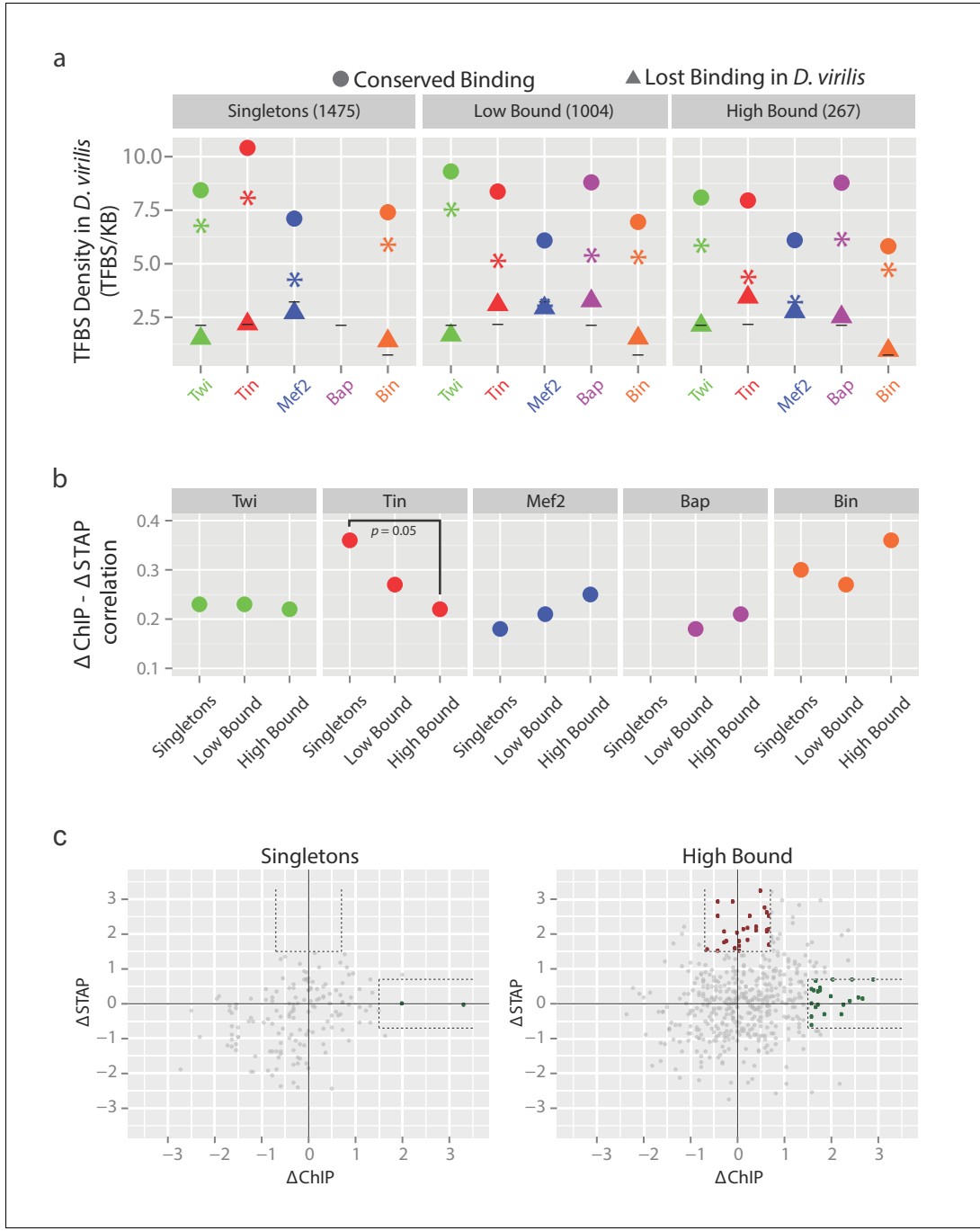

**Figure 4.** The relationship between TF binding conservation and motif conservation depends on enhancer context. (a) Orthologous CRMs between *D. melanogaster (D. mel)* and *D. virilis (D. vir)* were divided into 3 categories based on their occupancy in *D. mel*; CRMs bound by 1 TF (Singletons), 2 or 3 TFs (Low Bound) or by 4 or 5 TFs (High Bound). CRMs in each category were divided into CRMs with conserved occupancy (circle) and non-conserved occupancy (triangle) in *D. vir* and the density of transcription factor binding sites (TFBS) for the corresponding factor in *D. vir* was calculated (Materials and methods). Asterisks depict the differences in TFBS density between CRMs with conserved and non-conserved binding, while horizontal lines indicate average densities across the genome for comparison. The lack of conserved binding for Bap singleton CRMs results in the missing value for conserved Bap TFBS (also in [b]). (b) Spearman correlation between interspecies variation in motif presence (ΔSTAP) and variation in TF occupancy (ΔChIP) at orthologous CRMs for each TF, in the different occupancy classes. A high Spearman correlation indicates inter-species changes in TF binding due to inter-species changes in the presence of the TF's motif (i.e. binding changes are highly sequence-dependent). (c) Relationship

*Figure 4 continued on next page*

*Figure 4 continued*

between scaled *ΔSTAP* and *ΔChIP* for Singletons (left, 150 peaks) and High Bound (right, 488 peaks) orthologous enhancer pairs bound by Tin in *D. melanogaster*. Highlighted in red (0 and 28 peaks for Singletons and High Bound, respectively) are enhancer pairs with little changes in Tin occupancy (low *ΔChIP*) but high changes in Tin motifs (*ΔSTAP*) and in green (2 and 21 peaks for Singletons and High Bound, respectively) are enhancer pairs with high changes in Tin occupancy (high *ΔChIP*) with little changes in their Tin motifs (*ΔSTAP*). The former (red) are consistent with TF collective occupancy, while the latter (green) are consistent with Tin cooperative recruitment by other partner TFs. Dashed lines delineate the thresholds used. Note, many orthologous enhancers fulfill these criteria in the high-bound enhancers while very few or none do in the singletons class (p=0.06 and 0.0005 comparing (High Bound greater than Singletons) green and red populations (one sided Fisher test), respectively).

concomitant with loss of functional motifs for that TF (*Figure 4a*). This trend generally holds true for all three enhancer categories (High bound, low bound or singletons, *Figure 4a*), with the interesting exception of Tin, a TF previously shown to act in a TF collective at cardioblast enhancers (*Junion et al., 2012*). The number of Tin, and to a lesser extent Mef2 and Bin, motifs (overall motif density) in enhancers with conserved binding (*Figure 4a*, circles) tends to decrease as TF binding complexity increases (for example 7.96 versus 10.41 Tin motifs/kb (p-value < 5e-7, binomial test) in high-bound versus singleton enhancers, respectively) suggesting looser sequence requirements when the TF binds to regulatory elements collectively with other factors.

We confirmed this relationship between TF binding and sequence fitness with a threshold-free approach using STAP (sequence to affinity prediction) (*Cheng et al., 2013*), which integrates one or more strong or weak sites present in a sequence to produce a numeric prediction of TF occupancy in that sequence. STAP models were trained to discriminate between TF bound regions (significant ChIP peaks) and random sequences for each motif and in each species (Materials and methods). We then used the STAP-predicted TF occupancy to score ('STAP score') motif presence within each enhancer (Materials and methods, *Supplementary file 8*). For each orthologous enhancer pair, we calculated the interspecies differences between STAP score for each TF (*ΔSTAP*, indicating a change in predictive motifs) and the interspecies ChIP score (*ΔChIP*, indicating an observed evolutionary change in TF occupancy). This identified a positive and significant correlation between global *ΔSTAP* and *ΔChIP* for each TF (*Supplementary file 9*) indicating that divergence in TF binding is generally associated with a loss in that TF's motif, as expected. Repeating the *ΔSTAP-ΔChIP* analysis for each enhancer category separately (high-bound, low-bound and singletons, *Figure 4b*) showed a significantly lower correlation between TF occupancy predicted from sequence (*ΔSTAP*) and that observed (*ΔChIP*) in highly-bound CRMs compared to singleton CRMs for Tin (Fisher z-transformation, p=0.05, *Figure 4b*). Therefore, although Tin enhancer occupancy is generally deeply conserved in both cases (high or low bound CRMs), its underlying requirements for the Tin motif is reduced when the TF is bound with other factors. To investigate this further, we directly compared the observed evolutionary changes in Tin binding (*ΔChIP*) to changes in predictive Tin motifs (*ΔSTAP*) for orthologous enhancer pairs bound by Tin alone (singletons) or with many other TFs (High bound) (*Figure 4c*, Materials and methods). The scatter plot (*Figure 4c*) reveals two types of 'partner' relationships: (1) orthologous enhancers in which Tin binding has diverged (high *ΔChIP*), while the predictive motifs for Tin binding are conserved (low *ΔSTAP*) (*Figure 4c*, green dots) and (2) orthologous enhancers in which Tin binding is conserved (low *ΔChIP*), while the sequence fitness for Tin binding has diverged (high *ΔSTAP*) (*Figure 4c*, red dots). The former is consistent with a second partner TF being required to co-operatively recruit Tin to those enhancers, while the latter is consistent with the 'TF collective' model of enhancer occupancy (*Junion et al., 2012*), which proposed that enhancers bound by many functionally related TFs should be able to withstand some motif divergence while having little effect on the occupancy of the TFs themselves. Our cross-species occupancy data provides the first evidence that this prediction holds true for a collection of elements: in the context of enhancers bound by many TFs (4–5 factors), Tin occupies the enhancer in both species despite divergence in its motif, which is not the case when Tin binds alone (*Figure 4a–c*).

# Co-divergence in TF occupancy identifies potential cooperative binding of TFs

To identify potential mesodermal partner TF(s) that either preserve Tin binding across species despite divergence in its motif and/or are required to cooperatively recruit Tin, we assessed co-association between the occupancy of all TFs across all 14 conditions in both species. For this we determined if TF co-occupancy deviates from random expectations by deriving z-scores based on a background distribution generated by permuting the TF binding matrix 500 times (*Figure 5a*, Materials and methods). Indeed, Tin co-associates significantly with three other factors, Twi, Bap and Mef2, and to a lesser extent with Bin. Importantly, these co-associations in binding are present in both species, and therefore maintained across a large evolutionary distance, suggesting that the general properties by which these TFs regulate enhancer activity, and by extension the overall topology of the mesodermal gene regulatory network, is largely conserved.

If Tin binding depends on the binding of other TFs (*Figure 5a*), either directly or indirectly, then the loss of binding of this 'partner' TF during evolution should lead to loss of Tin binding at those orthologous enhancers. To assess this, we examined co-divergence in TF occupancy across species for all factors. Previous studies assessed similar TF partner (or combinatorial) relationships to explain loss of TF binding by searching for motif divergence for other TFs (*Borneman et al., 2007*; *He et al., 2011*; *Zheng et al., 2010*), or TF occupancy divergence (*Stefflova et al., 2013*). Here, we can use our measured co-divergence in TF occupancy to directly assess if the observed co-associations of Tin with other TFs are functionally required for TF occupancy.

Orthologous CRM pairs bound by two (or more) TFs in *D. melanogaster* were divided into all four possible binding categories: (1) CRMs with conserved binding of both TFs in *D. virilis*, (2) CRMs that have lost the binding of one TF (factor 1), (3) CRMs that have lost the binding of the second TF (factor 2) and (4) CRMs that have lost the binding of both TFs (*Figure 5b*). For each of the 10 possible co-association pairs (combinations of any two factors from the set of 5), we asked if the loss of both TFs in *D. virilis* is more frequent than expected, under the hypothesis of binding independence of the two TFs. Four TF pairs show such a significant binding dependency, which we refer to as co-divergence (*Figure 5b*, highlighted cells). In concordance with the association analysis (*Figure 5a*), three of the four cases involve Tin, and include Tin/Bap, Tin/Twi, and Tin/Mef2 (corrected p-values of $6 \times 10^{-11}$, 0.007, and 0.0006 respectively, *Figure 5b,d*). We next assessed if the loss of TF1 occupancy (*Figure 5c*, rows) is associated with sequence divergence (change in TFBSs) for TF2 (*Figure 5c*, columns) using STAP to score for presence of the TF2 motif. Consistent with the findings above, Tin-Bap emerged as the strongest associations (p=$2.09E^{-04}$, *Figure 5c*). Tin and Bap were suggested to bind as a heterodimer to one enhancer in *D. melanogaster* (*Zaffran and Frasch, 2002*). Our results indicate that both TFs have conserved co-binding to hundreds of elements (327 CRMs) during visceral mesoderm specification. This analysis also indicates the direction of dependency: Loss of Tin and Bap occupancy is associated with the loss of Bap and Tin motifs, respectively (*Figure 5c*, p=$2.09E^{-04}$ and p=$8.44E^{-04}$, respectively), a bidirectional dependency consistent with cooperative recruitment rather than via a heterodimer motif. As Tin directly regulates *bap* expression, this represents a conserved feed forward loop on these 327 developmental enhancers. Our results also suggest a bidirectional dependency between Tin and Twi occupancy (*Figure 5b,c*), suggesting that they act cooperatively to regulate early mesoderm enhancers. Mef2 and Tin exhibit a unidirectional association where the loss of Mef2 occupancy is associated with the loss of Tin motifs (p=$1.96E^{-04}$) but not vice versa (p=$8.05E^{-01}$)(*Figure 5c*). The last case of suggested dependent binding is Bap recruitment being dependent on Bin motifs (p=$3.71E^{-02}$, *Figure 5c*), two TFs essential for trunk visceral mesoderm development (*Jakobsen et al., 2007*; *Zaffran et al., 2001*).

Taken together, the co-association and co-divergence analysis indicate that the TFs in each of these pairs co-bind to enhancers in both species more frequently than expected, and that this co-binding appears to be required at some enhancers to preserve occupancy as when the binding of one factor is lost during evolution it is associated with the loss of binding of the second TF. The presence of Tin in the majority of co-divergent pairs suggests that the occupancy of Tin is highly cooperative, which could involve direct cooperativity that is dependent on the interplay between protein:: DNA and protein::protein interactions, although we note this could also involve indirect co-operativity through co-factor recruitment or nucleosome displacement.

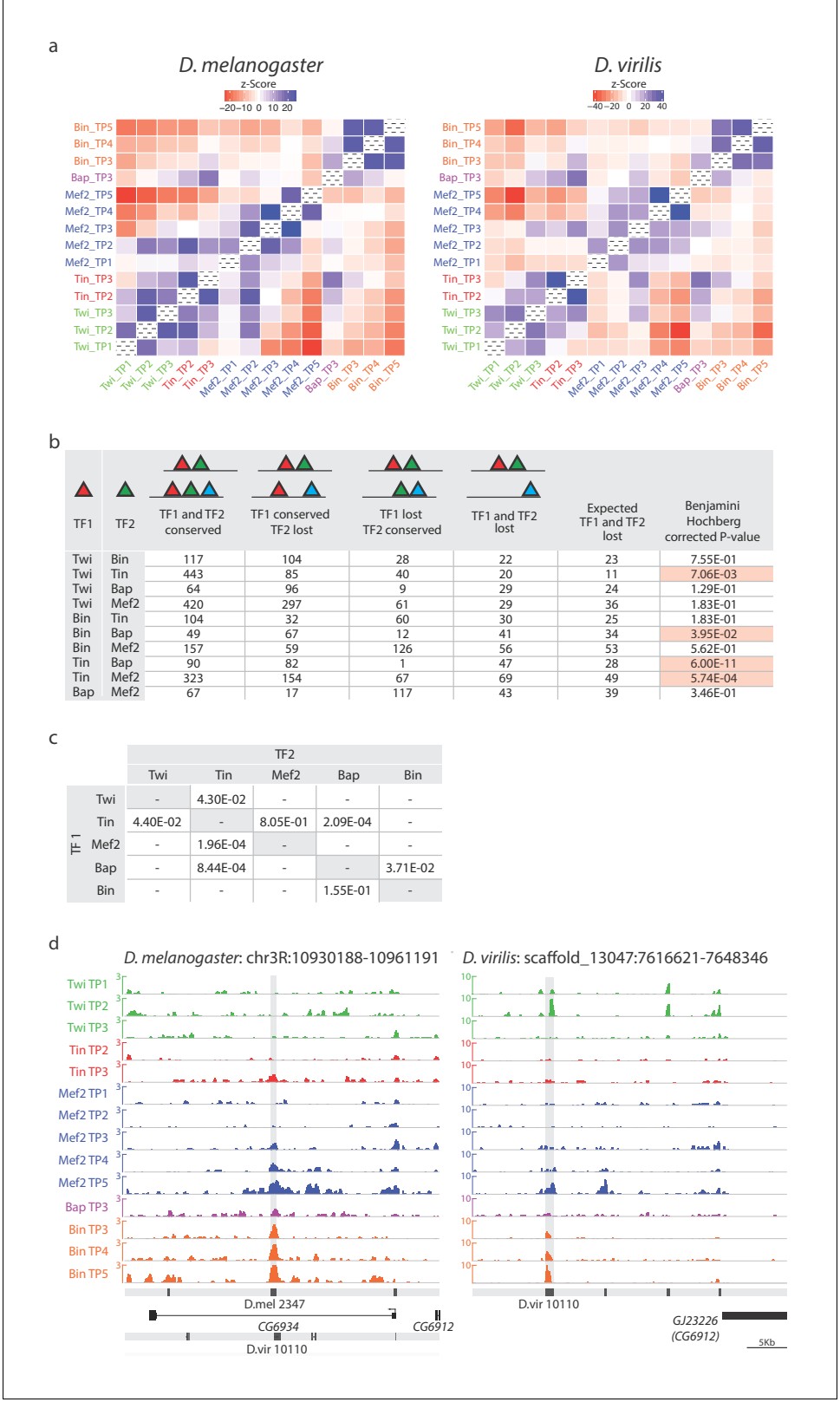

**Figure 5.** Co-divergence in TF occupancy across evolution can reveal potential cooperative interactions. (a) Pair-wise co-association of TF/time point in *D. melanogaster* (left) and *D. virilis* (right). Co-association significances correspond to Z-Score denoting deviation from random expectation ranging from red (lower than random expectation denoting lack of co-association) to blue (higher than random expectation denoting strong co-

*Figure 5 continued on next page*

*Figure 5 continued*

association). (**b**) Above table, schematic representation of the 4 categories of binding events affecting orthologous CRM pairs. Below, table indicates the number of CRM pairs in each category with p-values (Fisher exact tests). Significant cases (p<=0.05) are highlighted in red. (**c**) Association test between the loss of TF1 occupancy (TF1 ChIP, rows) and the loss of TF2 motifs (TF2 STAP, column). Significant associations between pairs are indicated (Fisher exact tests). (**d**) Occupancy of orthologous enhancers with co-divergent binding. Binding of both Tin and Bap are lost in *D. vir* (right) compared to *D. mel* (left). Track descriptions are as in *Figure 2c*. Orthologous CRM region is highlighted in gray.

## Linking evolutionary changes in TF occupancy to their effects on enhancer activity

The relationship between TF occupancy and enhancer activity is not straightforward. Enhancers occupied by similar combinations of TFs can regulate very similar patterns of expression (*Frankel et al., 2010*; *Hong et al., 2008*; *Perry et al., 2010*, *2011*). However, different combinations of TFs can also regulate very similar patterns of enhancer activity (*Brown et al., 2007*; *Liberman and Stathopoulos, 2009*; *Zinzen et al., 2009*). Over evolutionary time-scales, it is therefore not clear what effect changes in TF occupancy (enhancer input) will have on enhancer activity (enhancer output). A number of studies have indicated very good correlation between highly conserved binding and the expression of the closest proximal gene (*Ballester et al., 2014*; *Paris et al., 2013*). However, partial conservation of binding events within enhancers is much more common when distantly related species are compared, as observed here: almost 50% (1130/2442) of our orthologous multi-peak CRMs have partial conservation in the combination of TFs they recruit (*Figure 2a* and *Figure 3*). In these cases, it is difficult to predict what impact a moderate change in combinatorial TF binding (partial gain or loss in enhancer occupancy) will have on enhancer activity. Moreover, using the expression of the closest proximal gene as a proxy for enhancer activity is problematic for two reasons: First, a gene's expression represents the combined activity of multiple enhancers (often five or more), making it difficult to disentangle the functional consequences of evolutionary changes in TF occupancy at each individual element. Second, many enhancers do not regulate the closest proximal gene, but rather skip one or more genes, sometimes spanning large distances even in *Drosophila* (*Ghavi-Helm et al., 2014*).

To directly assess if the activity of orthologous CRMs is robust to changes in TF occupancy, we examined the extent to which conservation or divergence in TF occupancy, a criteria widely used to infer functional importance of TF binding events, are consistent with conserved or diverged enhancer activity. For this, we took advantage of our previous predictions of the activity of the 8008 *D. melanogaster* CRMs used here (*Zinzen et al., 2009*), in which we demonstrated that Support Vector Machines (SVMs) trained with the binding signatures of TFs on characterized enhancers could qualitatively predict enhancer activity with high accuracy in four tissue classes (*Zinzen et al., 2009*). When the predictions were tested in-vivo using transgenic embryos, the enhancers were active in the predicted tissue in ~85% of cases (*Cannavò et al., 2016*; *Zinzen et al., 2009*). Here we compared the spatio-temporal expression patterns of both *D. melanogaster* and *D. virilis* orthologous CRM pairs (*Figure 6*). For this, we selected pairs with medium to high divergence in TF binding to orthologous CRMs (i.e. a medium to low Jaccard index (J <= 0.5); *Supplementary file 10*) in addition to having a high SVM specificity (spec. >0.98) of predicted expression for the *D. melanogaster* CRM. Of the 161 orthologous pairs passing these criteria, we selected seven pairs after manual checking for a good alignment between orthologous loci.

All 14 CRMs (7 *D. melanogaster* and their 7 *D. virilis* orthologs) were assayed in transgenic reporter assays in *D. melanogaster* (Materials and methods, *Supplementary file 10*). Five CRM pairs (71%) showed conserved tissue activity between both species (*Figure 6a*), of which four are fully concordant with the SVM tissue predictions and one has partial concordance predicting one tissue correctly (*Figure 6a*). The fifth pair (*D. mel* CRM-4682/*D. vir* CRM-12291) has conserved activity in the mesoderm with additional activity in *D. virilis* in two mesodermally derived tissues, the visceral muscle (VM) and somatic muscle (SM) (*Figure 6a*, *Figure 6—figure supplement 1*). We note that three CRM pairs with conserved overlapping activity have activity in other non-overlapping tissues indicating that these elements are also bound by other TFs not assessed here (*D. mel* CRM-6087/*D. vir*

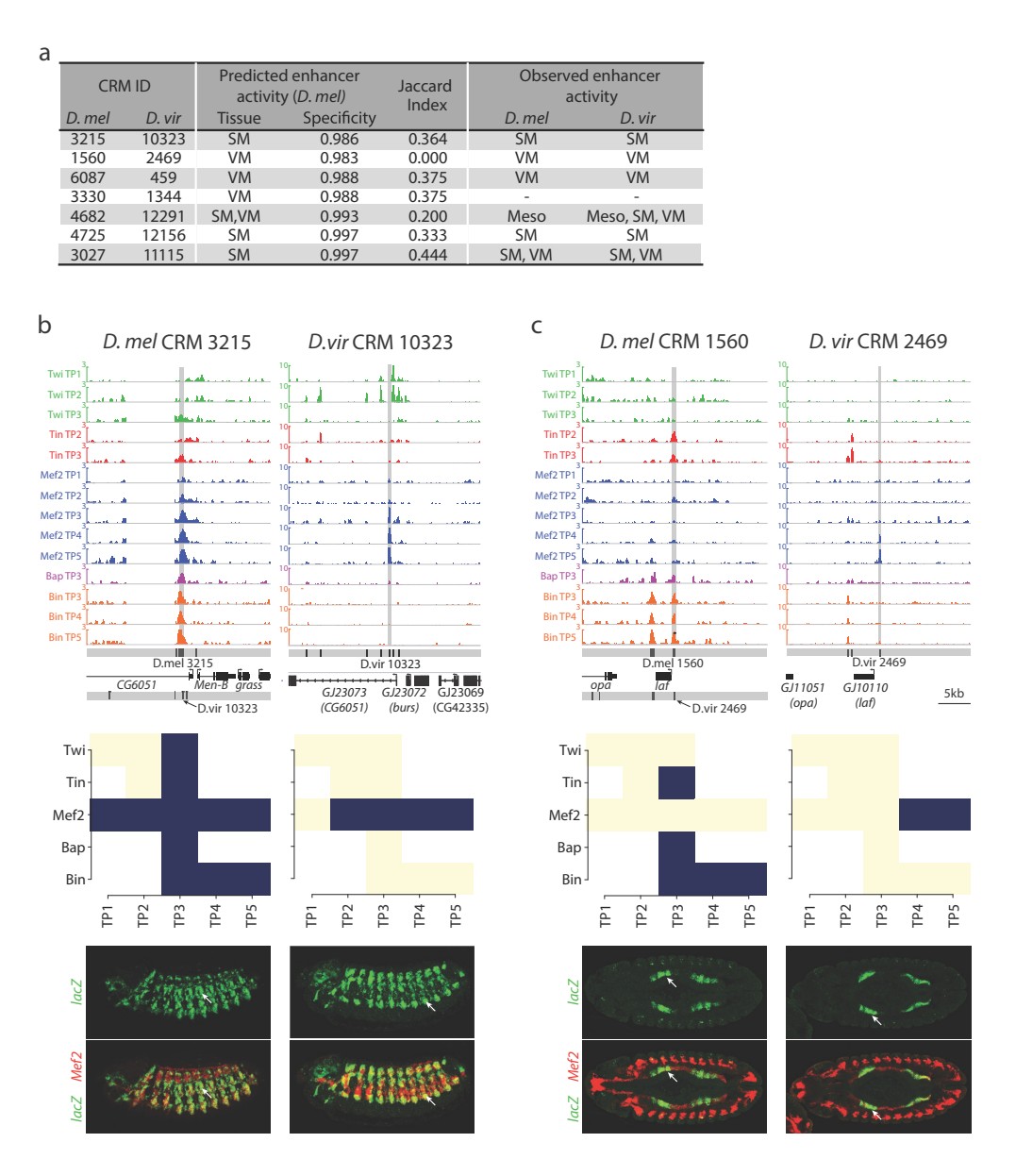

**Figure 6.** Relationship between divergence in TF occupancy and enhancer activity. (**a**) Table summarizing the orthologous *D. melanogaster* (*D. mel*) and *D. virilis* (*D. vir*) CRMs tested for in-vivo activity in *D. mel* embryos. The enhancer's predicted tissue activity, and SVM specificity score, for the *D. mel* enhancer is indicated (from **Zinzen et al., 2009**). The observed enhancer activity for both the *D. mel* and orthologous *D. vir* enhancers are indicated. SM = somatic muscle, VM = visceral muscle, Meso = mesoderm. (**b–c**) Upper panel, TF occupancy (ChIP signal) of two orthologous CRM pairs representing the quantitative binding profiles along the 14 conditions in both species. *D. mel* data corresponds to log2(IP/mock) ChIP-chip signal. *D. vir* data corresponds to RPM normalized, input subtracted ChIP-Seq signal. In *D. mel*, both *D. mel* and translated *D. vir* CRMs are shown along with *D. mel* gene models. In *D. vir*, *D. vir* CRMs and *D. mel* orthologous gene models are shown. Middle panel shows TF binding logos, where blue represents significant ChIP peaks at the indicated time point (columns = Time Point (TP) and rows = factor). Lower panel shows double in-situ hybridization of transgenic embryos for each orthologous enhancer (*lacZ* reporter driven by the enhancer, green channel) and a mesodermal marker (*Mef2*, red). White arrow indicates the somatic muscle (**b**) and visceral muscle (**c**) of an embryo in a lateral or dorsal view, respectively.

The following figure supplement is available for figure 6:

**Figure supplement 1.** Binding logo, in vivo data and browser screenshots for tested CRMs with mesodermal activity.

CRM-459; *D. mel* CRM-4682/*D. vir* CRM-12291; *D. mel* CRM-4725/*D. vir* CRM-12156). The last pair (*D. mel* CRM-3330/*D. vir* CRM-1344), with a *D. melanogaster* SVM prediction in the VM, showed no activity in either species despite conserved Bin binding.

Overall, the five orthologous enhancers with conserved tissue activity (output) have extensive divergence in their TF occupancy, with less than 40% of their TF binding conserved (*Figure 6a*, Median Jaccard index = 0.37). For example, *D. mel* CRM-3215 is bound by all five TFs in *D. melanogaster* while in *D. virilis* it is only bound by Mef2 (*Figure 6b*). Even more striking is *D. mel* CRM-1560, which is bound by Tin, Bap and Bin in melanogaster (*Figure 6c*). All significant binding for these factors is lost in *D. virilis* (CRM 2469), but the enhancer has gained Mef2 binding – yet the very specific VM activity is conserved (Jaccard index = 0, *Figure 6a,c* and *Figure 6—figure supplement 1*).

These data indicate that divergence in TF binding does not necessarily equate to divergence in enhancer activity, as generally assumed. The strong conservation in orthologous enhancer output, despite a low conservation in TF binding may reflect stabilizing selection acting to maintain the correct spatio-temporal activity of orthologous developmental enhancers.

## Discussion

The evolutionary properties of TF function can be considered at two levels; (1) the relationship between sequence divergence and TF occupancy and (2) the relationship between TF occupancy divergence and enhancer activity. While many studies have addressed the first, few have tackled the second, and when they have, they focused on an individual well-characterized enhancer. This study touches on both aspects, where we look at global properties of TF occupancy within the context of orthologous developmental *cis*-regulatory elements.

We initiated this study by measuring the cross-species occupancy of 5 developmental TFs, which function within the same gene regulatory network to specify the subdivision of the mesoderm into different muscle primordia. We used this data to assess the impact of divergence in DNA sequence and TF occupancy at the level of *cis*-regulatory elements, rather than individual binding events, as these are the functional units through which selection most likely acts.

### The relationship between sequence divergence and TF occupancy

Divergence in TF occupancy is often related to either (1) loss of the binding site for the TF itself (*He et al., 2011*; *Kasowski et al., 2010*; *Schmidt et al., 2010*; *Zheng et al., 2010*) or (2) the loss of TFBSs for additional partner TF (*Paris et al., 2013*; *Stefflova et al., 2013*), as depicted in *Figure 7* (upper panels). Our study confirms these findings, showing that loss of TF binding is associated with either the loss of the TF's motif (*Figure 4*) or the loss in TF binding of an associated 'partner' TF (due to the loss of its motif) that combinatorially binds to the same element (*Figure 5*). We also observed the converse – conserved TF binding despite sequence changes, that is, divergence of that TF's binding site (*Figure 4c* and schematized in *Figure 7*, lower panels). This property depends on the degree to which the TF is combinatorially bound to an enhancer with other TFs (*Figure 4b,c*). When a TF co-occupies an enhancer with many other functionally related TFs (i.e. as a TF collective [*Junion et al., 2012*; *Spitz and Furlong, 2012*]) its occupancy is less sequence dependent. For example, when Tin is bound to enhancers with few other mesodermal TFs, the conservation in its binding is correlated with conservation in its TFBS. However, when Tin is bound to enhancers that are highly bound by other mesodermal TFs, for whom Tin is known to function with, its conservation in binding is less dependent on its DNA motif (*Figure 7*). In other words, the collective binding of Tin with other mesodermal TFs can compensate for divergence in its motif, in keeping with the predicted increased flexibility proposed by the TF collective model (*Junion et al., 2012*; *Spitz and Furlong, 2012*; *Uhl et al., 2016*) (*Figure 7*).

### The relationship between divergence in TF occupancy and enhancer activity

The 'input – output' relationship of enhancers, a trait on which selection directly acts, has been largely unexplored in global comparative cross-species studies of TF binding. The large evolutionary distance between *Drosophila melanogaster* and *virilis* (equivalent to that between humans and lizards based on neutral mutation rate [*Stark et al., 2007*]), has led to extensive motif changes within

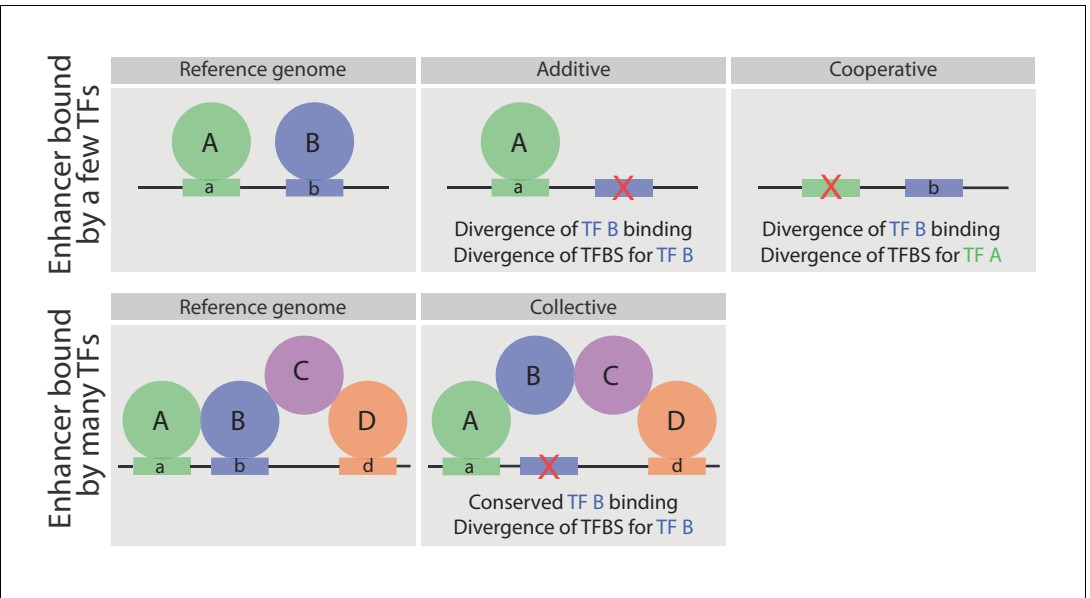

**Figure 7.** Schematic representation of the relationship between sequence divergence and TF occupancy on enhancer elements. Upper panel, indicates an enhancer bound directly by two factors, A and B. In some enhancers the loss of TF binding (e.g. TF B) is associated with the loss of that TF's binding site (motif b - middle). In other enhancers (right), the loss of a partner TF's motif (motif a) will result in the loss of binding of TF B, indicating that TF B occupies the enhancer cooperatively (either directly or indirectly) with TF A. Both cases have been observed in inter- and intra- species analysis of TF occupancy, and are also observed here with these mesodermal developmental TFs. Lower panel, represents an enhancer bound by a TF collective. Here, the loss of motif b does not result in loss of TF B binding (middle panel), unlike the situation in enhancers with purely additive occupancy (depicted above). Rather the divergence in the TF binding site is tolerated as the TF's binding is stabilized through protein-protein interactions.

these orthologous enhancers. We used this high sequence divergence to tease apart functional associations between TFs, identifying potentially cooperative pairs of TFs whose co-associations are essential for TF recruitment at enhancers. Importantly for evolutionary analyses, these cooperative interactions often allow a TF to occupy an enhancer (conserved binding) despite a loss of its functional motif (diverged sequence), as measured using STAP, a threshold free thermodynamics-based model of quantitative occupancy (*Figure 5c* and *Supplementary file 9*). This suggests that combinatorial binding may offer evolution a greater neutral neighborhood, and thus can increase the robustness of developmental systems.

When associating TF occupancy with enhancer activity it is clear that not all TF binding events behave equally; the loss or gain of TF binding at an enhancer during evolution can have diverse effects. By examining the activity of 7 orthologous enhancer pairs with high divergence in their TF occupancy in transgenic embryos, we uncovered extensive 'plasticity' of enhancers to evolutionary changes in TF occupancy. Despite highly diverged TF occupancy (enhancer input), their tissue activity (output) was often highly conserved. Therefore, a loss in TF binding does not necessarily lead to a loss (or even a tissue change) in an enhancer's activity, rather it is an inherent property of the enhancer due to the combination of regulatory inputs that define its activity.

## Conservation in TF occupancy in relation to the TF's function and position within a developmental GRN

Our results suggest a potential relationship between the extent of conservation in TF binding and the position of the TF within its developmental gene regulatory network. Four of these TFs directly regulate each-others expression in a linear cascade (*Figure 1a*), providing an excellent system to examine this: Twi directly regulates *tin* expression (*Yin et al., 1997*), Tin regulates *bap* (*Lee and Frasch, 2005*) and Bap in turn regulates *bin* expression (*Zaffran et al., 2001*). The TFs higher up in

the network (Twi and Tin) generally have more conserved occupancy (~54%) compared to TFs lower down in the network (Bap and Bin; 19% and 25% respectively) (*Figure 1a*, *Supplementary file 1*). Interestingly, studies, in diverse developmental processes and species have observed both cases; higher variation in key regulators towards the bottom of the GRN in line with our findings (*Abzhanov et al., 2006*; *Garfield et al., 2013*), or conversely at the top of their respective GRN (*Hinman et al., 2007*, *2003*). The extent of divergence and conservation of TF activity may therefore depend on the function of the GRN itself (*Erwin and Davidson, 2009*).

The TFs examined in this study have a conserved role in mesoderm specification and muscle development from flies to man, and form part of an evolutionarily conserved GRN. Our results indicate that the evolutionary patterns acting on this network are shaped by the collective interactions between these TFs, as much as the functions of any individual factor themselves. Evolutionary divergence of an individual TF should therefore not be assessed in isolation. It is rather the context in which developmental TFs act that determines their evolution, both at the enhancer (i.e. the presence of other TFs at each individual enhancer) and network (i.e. the position of a TF within a GRN and its role in feed forward and feedback regulation) levels.

## Materials and methods

### D. virilis embryo staging

*Drosophila virilis* (w[-]) was reared on standard bottles, identical to *D. melanogaster*. To establish a large population of *D. virilis*, we found extra feeding at day 3 if reared at 25 degrees and day 5 if reared at 18 degrees critical to efficiently amplify the stock. To determine equivalent stages between the two species, rounds of small-scale timed embryo collections were obtained at 25°C for *D. virilis* embryos to define time intervals with the desired stages. Following each collection, embryos were fixed and immunostained using anti-*D. melanogaster* Mef2 to precisely stage each embryo. The distribution of stages was manually counted for each sample's embryo collection using a Zeiss LSM 510 META confocal microscope.

### ChIP-seq on D. virilis embryos

Staged *D. virilis* embryo collections, at the defined time intervals (*Figure 1a*), were obtained at 25°C and fixed in 1.8% formaldehyde for 15 min. Extracted chromatin was sonicated on a Bioruptor for 18 cycles of 30 s ON, 30 s OFF at high frequency, yielding an average shearing size of 250–300 bp. Rabbit polyclonal antibodies targeting the full-length protein for *D. virilis* Twi (2 independent sera), Tin, Bap and Bin were generated specifically for this study to increase target specificity. For this, the protein was expressed in, and purified from, *E. coli* as a SUMO fusion protein and used as antigen to raise the antibodies. The SUMO tag was used for protein purification. *D. melanogaster* anti-Mef2 was highly specific when tested on *D. virilis* embryos (both by immunostaining (*Figure 1—figure supplement 1*) and ChIP) and thus used for Mef2 *D. virilis* ChIP.

Chromatin from *D. virilis* staged embryos was extracted as described previously for *D. melanogaster* (*Sandmann et al., 2006*) and ChIP conditions were optimized for each TF by ChIP-qPCR using the recovery of positive and negative regions (*Supplementary file 12*). All antibodies were first pre-absorbed using chromatin from 0 to 2 hr *D. virilis* embryos, a time window when none of the TFs are expressed. Two independent biological replicates were generated for each transcription factor at each time-point, using two independent antibodies when possible (for Twi and Mef2, targeting the full-length and DNA binding domains, respectively). Although we have no guarantee that both sera target different portions of the protein, these polyclonals were generated in different rabbits and therefore have a different background level of IgG antibodies in the sera. Immunoprecipitated DNA was used for library preparation, and sequenced by multiplexing 10 samples on one Illumina HiSeq lane. Two genomic input samples were sequenced for each time point.

### ChIP-seq sample processing

Reads were aligned using Bowtie 1 (-n 2 -e 70 l 28 –maxbts 800 -y -m 1 –best –strata -S –phred33-quals) (*Langmead et al., 2009*) on Flybase-R1.2 assembly version of the *D. virilis* genome (corresponds to droVir3 in UCSC) and filtered for PCR and optical duplicates using 'rmdup' function of Samtools (*Li et al., 2009*). Peak calling was performed using MACS2 (-g 1.2e8 –to-large and -p 1e-3

as requested for the IDR analysis, see below) against stage-matched input. MACS2 peaks were selected using the Irreproducibility Discovery Rate (IDR) measure with the workflow (*Landt et al., 2012*) using a 1% IDR threshold leading to a unique, highly confident and consistent peak sets for biological replicates. More precisely, we selected the top Np peaks from MACS2 ran on merged replicates (where Np is the number of peaks returned at a 1% IDR threshold on the pooled pseudo-replicates analysis, see *Figure 7* of [*Landt et al., 2012*]). Note that the selected peaks have MACS2 q-values < 0.001 in all conditions. For visualization, BAM file reads were extended to the average length of the genomic fragments for the corresponding experiment, merged and scaled to Read Per Million (RPM) followed by input subtraction of the control from the corresponding time point using deeptools (*Ramírez et al., 2014*).

Motif discovery on ChIP-seq samples was done using the console version of MEME-ChIP with the parameters '-meme-mod zoops -meme-nmotifs 5 -meme-minw 6 -meme-maxw 15' (*Machanick and Bailey, 2011*). For each sample, fasta sequences of IDR defined peak summits were extended by 100 bp on 3' and 5'and used as input. Discovered Position Weight Matrices (PWM) are shown in *Figure 1—figure supplement 2*.

## Defining ChIP CRMs across two distantly related species

*D. virilis* ChIP-seq peak summits across all conditions were clustered using Bedtools (Quinlan and Hall) with a maximum distance of 400 bp between adjacent summits. Clusters were then extended by 100 bp to account for possible uncertainty in defining peak summits with high resolution. Coordinates of defined CRMs are provided in *Supplementary file 2*.

To identify orthologous TF binding events *D. virilis* IDR based MACS2 Peaks and CRMs coordinates were translated to *D. melanogaster* using the pslMap, an implementation of the TransMap alignment algorithm (*Zhu et al., 2007*). pslMap uses a precomputed alignment and assigns orthologous information based on alignment scores in a base-by-base manner, taking rearrangements including insertions and deletions into consideration, thereby allowing expansion or contraction of block sizes. pslMap is therefore more suitable for translation between species, while liftover is more suitable for translation between assemblies. Upon translation, 10,532 *D. virilis* CRMs (73% of the total 14,385 CRMs) were uniquely mapped. 275 CRMs mapped to more than one locus in *D. melanogaster,* yielding 541 CRMs (10,532 + 541 = 11,073).

The 10,532 *D. virilis* CRMs were overlapped with the 8,008 *D. melanogaster* CRMs to define a high confidence set of orthologous putative developmental enhancers. Clusters of overlapping CRMs involving >1 CRMs of either species were filtered out, resulting in a conservative set of 2846 orthologous CRM pairs. For TFBS density analysis (*Figure 4a*), each orthologous CRM of a pair was extended to an identical size S, equal to the sum of the size of both pair of CRMs (S/2 bases was added on each side of the CRM center). TFBS density analysis for each TF over all TFBS densities (*Figure 4a*) were computed using TFBSs located within (1) a 100 bp region around TF binding peak summit(s) when the CRM is bound by the considered TF (conserved binding) or (2) a 200 bp window randomly selected within the CRM boundaries when the CRM is not bound by the considered TF (lost binding).

## Enrichments of characterized enhancers

To assess if our conserved set of CRMs are enriched for known mesodermal enhancers, we extended our previously built CRM Activity Database (CAD [*Zinzen et al., 2009*]) by adding the activity information of active tiles from Kvon and colleagues (*Kvon et al., 2014*) and a set of new entries from RedFly (*Gallo et al., 2011*). This led to a final set of 8876 enhancers of which 1254 are active in the mesoderm and 2970 are active in other tissues, excluding the mesoderm (*Supplementary file 7*). The remaining 3105 regions show no or ubiquitous staining. Fisher exact test was used to assess the significance of overlaps between different categories.

## Hierarchical clustering of ChIP quantitative signal

Heatmaps for clustering of CRM pairs in conservation categories were constructed using the log2 (IP/control) for both *D. melanogaster* ChIP-chip and *D. virilis* ChIP-seq data. The Ward algorithm was used to cluster ChIP signals across orthologous CRMs using Pearson correlation to calculate distance. The R package 'pheatmap' was used to draw the heatmaps at a fixed scale for comparison.

## Differential motif analysis on temporally bound interspecies enhancers

Early and late bound enhancers were determined in each species separately. First, enhancers only bound early ('early enhancers') were required to have a Twi ChIP peak at 2–4 hr (2–5 hr in *D. virilis*) but not at later stages while enhancers only bound late ('late enhancers') were required to have a Twi ChIP peak at 6–8 hr (7–10 hr in *D. virilis*) but not at earlier stages. Second, enhancers selected in the first step were post filtered using the quantitative ChIP signal, being required to have a ChIP signal ratio greater than two early/late or late/early for early and late enhancers, respectively. The final numbers of temporally bound elements were: 638 for Twi early, 439 for Twi late in *D. melanogaster* and 490 for Twi early and 254 for Twi late in *D. virilis*.

Motifs from the FlyFactorSurvey database (downloaded from the MEME (*Bailey et al., 2015*) web site) preferentially enriched in Twi early vs Twi late enhancers (and conversely) were identified in *D. melanogaster* and *D. virilis* separately using the following procedure. First, AME (*McLeay and Bailey, 2010*) was run with –scoring totalhits –method fisher –pvalue-threshold 0.0001 –pvalue-report-threshold 0.05 options on late bound enhancers with early bound enhancers as background, and conversely. Second, motif matches (for motifs reported by the AME analysis) were identified in enhancers using FIMO (*Grant et al., 2011*) ran with a similar p-value threshold (–thresh 0.0001) and 0-order background files generated using the genome fasta files. For each motif reported by AME, the overall ratio R (match number per Kb of sequence in early bound CRM/match number per Kb of sequence in late bound CRM) was computed and only motifs exhibiting an absolute log2(R) greater than 1 (i.e. 2 fold difference between early and late set) were considered in the heatmap presented in *Figure 3b* (redundant motifs were removed for clarity). The heat map therefore presents the (de-duplicated) motifs (1) found significant in either *D. melanogaster* or *D. virilis* AME analysis (or both) and (2) having an abs(log2(R))>1 in the organism where the motif was reported significant by AME.

## Transcription factor binding site predictions and thresholds

Transcription factor binding sites (TFBS) predictions for Twi, Mef2, Tin, Bap and Bin were generated using patser (*D. virilis* options: -A a:t 0.28 c:g 0.18 -d2 -c -ls 0 s; *D. melanogaster* options -A a:t 0.28 c:g 0.20 -d2 -c -ls 0 s) (*Hertz and Stormo, 1999*) in both *D. melanogaster* and *D. virilis* genomes. For each TF, we selected the best performing PWM between PWMs discovered using *D. virilis* ChIP-seq peaks (this study, see 'Motif discovery' section) and the *D. melanogaster* PWM derived from ChIP-chip (*Zinzen et al., 2009*). For each PWM, patser was run with very low stringency (-ls 0) to predict TFBSs genome-wide in both *D. melanogaster* and *D. virilis* genomes. Each PWM was then evaluated using ROC plots representing true positive and false positive results gained from the whole range of patser scores (0 and higher). For each species, ChIP CRMs were split into two groups: a group containing ChIP CRMs bound by the considered transcription factor and a group containing ChIP CRMs not bound by that transcription factor. The latter was used as 'background' regions to control for general sequence biases in ChIP CRMs, such as GC content. These CRM groups were then used to assemble the positive and negative sets as 200 bp regions centered on ChIP peak summits. Note that summits were as (1) predicted by MACS2 for ChIP-seq or (2) obtained from Zinzen and colleagues (*Zinzen et al., 2009*) and translated from dm2 to dm3 using UCSC lift-over for the positive set; while a random summit position was sampled within each unbound CRM for the negative sequence sets ('background'). Peaks from the positive and negative sets with a TFBS having a patser score $\geq$x were identified for x ranging from 0 to the maximum observed score, and used to build ROC plots using ROCR (*Sing et al., 2005*) presented on *Figure 1—figure supplement 2*. These plots indicate *D. melanogaster* Tin and Bin PWMs and *D. virilis* Mef2, Bap and Twi PWMs as the best performing models and are therefore used here. We finally determined the final thresholds as (1) the patser score giving a TPR of 70–80% and a FPR of 40% maximum on the best performing organism (as evaluated by the ROC) and (2) the patser score giving the same TPR for the other organism.

## STAP analysis

STAP (Sequence To Affinity Prediction) is a thermodynamics-based method for predicting TF occupancy in a DNA segment, given the TF's binding motif (*He et al., 2009*). We trained STAP models for each of the 14 conditions (TF, time point pairs) and in each species separately, following the method we used previously in *Cheng et al. (2013)*. Each training set contained the top 1000 ChIP

peaks and 1000 random windows of the same length, and their respective ChIP scores (suitably normalized, see below). Once the single free parameter in STAP was trained on these data, we used the trained model to score each CRM for motif presence. For each TF, the binding motif (PWM) was designated based on the best performing PWMs discovered from *D. melanogaster* and *D. virilis* ChIP data sets (this study, see 'ChIP-seq sample processing' section). To make the STAP scores comparable across species, we performed the following normalization on the ChIP scores in the training set: we set $\mu + 3\sigma$ as the maximum ChIP score, where $\mu$ and $\sigma$ are the mean and standard deviation of ChIP scores in the training set, replacing all ChIP scores greater than this maximum with $\mu + 3\sigma$, and finally applied min-max normalization to set all ChIP scores in a 0–1 range. We also normalized STAP scores computed by the trained model, using the same normalization procedure as above, separately for each data set.

To assess the accuracy of STAP on each of the 28 data sets (combinations of TF, time point, species), we applied 4-fold cross validation on the 2000 DNA segments mentioned above (each fold used 1500 segments to train the single free parameter in STAP and to score the remaining 500 segments). The resulting set of 2000 STAP scores were compared to respective ChIP scores, using Pearson correlation coefficient (PCC). The models fit the data very well, with PCC = 0.51 being the average over the 28 data sets (*Supplementary file 8*). For comparison, the average PCC in our previous analysis of 45 different ChIP data sets (also using 2000 segments) was ~0.30 (*Cheng et al., 2013*), and that the p-value of a PCC of 0.51 calculated on 2000 points is $<1e^{-133}$. *Figure 4b* shows the Spearman correlation between differences in the normalized interspecies ChIP scores, $\Delta$ChIP = ChIP$_{Dmel}$ - ChIP$_{Dvir}$ (indicating an observed evolutionary change in TF occupancy) and the differences in the normalized STAP scores, $\Delta$STAP = STAP$_{Dmel}$ - STAP$_{Dvir}$ (indicating a change in predictive motifs) for each orthologous enhancer pair.

*Figure 4c* shows this correlation (interspecies differences between the normalized $\Delta$STAP and $\Delta$ChIP scores) for the subset of elements that represents two orthologous enhancer pairs bound by Tin in *D. melanogaster*. $\Delta$ChIP and $\Delta$STAP scores were converted to Z scores using the *scale()* function in R. Scatter plots presented in *Figure 4c* show the $\Delta$STAP-$\Delta$ChIP correlation of Tin Singletons (150 elements bound only by Tin in *D. melanogaster*) and High Bound enhancers (488 elements bound by Tin plus 3 or 4 additional TFs in *D. melanogaster*). Highlighted enhancer pairs were selected using the following criteria: (1) low absolute $\Delta$STAP (abs($\Delta$STAP)<0.7) and high $\Delta$ChIP ($\Delta$ChIP > 1.5) correspond to orthologous enhancers in which Tin binding in *D. melanogaster* is largely reduced (or lost) in *D. virilis* without noticeable variation in the sequence fitness for Tin binding; and (2) high $\Delta$STAP ($\Delta$STAP > 1.5) and low absolute $\Delta$ChIP (abs($\Delta$ChIP)<0.7) correspond to orthologous enhancers in which Tin binding in *D. melanogaster* is not affected in *D. virilis* while the sequence fitness for Tin binding is largely reduced.

## Co-association and co-divergence analysis

Pairwise TF/time point co-association (*Figure 5a*) was performed following the method described by Cheng *et al.* with minor changes (*Cheng et al., 2014*). Briefly, we generated binary matrices with CRMs in rows and conditions in columns with cells containing 0 or 1 whether the CRM is bound at that condition or not, respectively. A real 14 × 14 association similarity matrix was generated first containing the number of times a certain association of conditions exists. We then permuted the content of each column and each row in the binary matrix by keeping the counts per column and row unchanged, and re-calculated a 'pseudo' association similarity matrix each time, for 500 times. This corresponds to the background distribution. The real association numbers for each pair of conditions were then compared to the 'pseudo' association numbers to derive z-scores.

To assess co-divergence in TF occupancy (*Figure 5b*), orthologous CRM pairs bound by two (or more) TFs in *D. melanogaster* were divided into all four possible binding categories: (1) CRMs with conserved binding of both TFs in *D. virilis* ('TF1 and TF2 conserved'), (2) CRMs that have lost the binding of one TF (factor 1, 'TF1 lost and TF2 conserved'), (3) CRMs that have lost the binding of the second TF (factor 2, 'TF1 conserved and TF2 lost') and (4) CRMs that have lost the binding of both TFs ('TF1 lost and TF2 lost'). For each of the 10 possible co-association pairs (combinations of any two factors from the set of 5), the table in *Figure 5b* presents the CRM numbers found in each of the four categories together with the p-value of the Fisher exact test (with Benjamini & Hochberg correction for multiple testing) evaluating the binding independence of the two TFs. The expected number of *D. virilis* CRMs that would have lost the binding of both TFs under the null hypothesis

('Expected TF1 lost and TF2 lost' column of *Figure 5b*) is computed as: $\text{Expected}_{1,2} = S_{1,2} * P_1 * P_2$ where $S_{1,2}$ is the number of orthologous CRMs co-bound by TF1 and TF2 in *D. melanogaster* (sum of the 'TF1 and TF2 conserved', 'TF1 lost and TF2 conserved', 'TF1 conserved and TF2 lost' and 'TF1 lost and TF2 lost' columns), $P_1$ is the observed probability of a *D. melanogaster* CRM to lose $TF_1$ binding in its *D. virilis* orthologous CRM (sum of 'TF1 lost and TF2 conserved' and 'TF1 lost and TF2 lost' divided by $S_{1,2}$) and $P_2$ is the observed probability of a *D. melanogaster* CRM to lose $TF_2$ binding in its *D. virilis* orthologous CRM (sum of 'TF1 conserved and TF2 lost' and 'TF1 lost and TF2 lost' divided by $S_{1,2}$).

## Transgenic reporter assays

To assay ChIP-CRMs for their enhancer activity, corresponding genomic regions were amplified by PCR (primers list in *Supplementary file 11*) and placed upstream of minimal *eve*-promoter driving a *lacZ* reporter gene. All constructs were injected according to standard methods (Rubin, G.M. and Spradling, A.C, *Science,* 1982) into the landing site line M{3xP3-RFP.attP'}ZH-51C (Basler lab, Kosman, D et al., *Science,* 2004), yielding integration at chromosomal position 51C. Transgenic lines were generated by BestGene Inc.

Subsequently, formaldehyde-fixed embryos were stained by double fluorescent in situ hybridization (Furlong et al., *Science,* 2001) using anti-sense probes against *lacZ* and *Mef2*. All images were taken with a Zeiss LSM 510 META confocal microscope.

## NGS data availability

Raw sequence data has been deposited in ArrayExpress under accession number E-MTAB-3798 (*D. virilis* Twi, Mef2, Tin, Bap and Bin ChIP-seq developmental time courses) and processed files for visualization are available at: http://furlonglab.embl.de/data/

## Acknowledgements

We are very grateful to all members of the Furlong lab for discussions and comments, especially David Garfield. This work was technically supported by the EMBL Genomics Core facility and the public resources of FlyBase, BDGP and RedFly. The work was financially supported by the Marie Curie EvoNet ITN and DFG FU 750 grants to EEF, NIH grant R01GM114341 to SS and an EMBO long-term fellowship 1125-2010 to PK.

## Additional information

### Funding

| Funder | Grant reference number | Author |
| --- | --- | --- |
| Marie Curie | ITN EvoNet | Pierre Khoueiry<br>Eileen EM Furlong |
| Deutsche Forschungsge-meinschaft | DFG FU750 | Eileen EM Furlong |
| National Institutes of Health | R01GM114341 | Saurabh Sinha |
| European Molecular Biology Organization | Long-term fellowship | Pierre Khoueiry |

The funders had no role in study design, data collection and interpretation, or the decision to submit the work for publication.

### Author contributions

PK, Conceptualization, Resources, Data curation, Supervision, Funding acquisition, Investigation, Writing—original draft, Project administration, Writing—review and editing; CG, LC, Resources, Data curation, Formal analysis, Investigation, Visualization, Methodology, Writing—original draft, Writing—review and editing; P-CP, Validation, Investigation, Visualization, Writing—review and editing; EHG, Formal analysis, Writing—review and editing; SS, Resources; EEMF, Software, Formal

analysis, Supervision, Funding acquisition, Investigation, Methodology, Writing—original draft, Writing—review and editing

## Author ORCIDs
Pierre Khoueiry, http://orcid.org/0000-0001-7643-3310
Charles Girardot, http://orcid.org/0000-0003-4301-3920
Eileen EM Furlong, http://orcid.org/0000-0002-9544-8339

## Additional files

### Supplementary files

• Supplementary file 1. Number of reads and peaks called in *D. virilis* and translated. Shown are number of reads used for both replicates, number of peaks called using IDR of 1% (Materials and methods) on *D. virilis* along with the number of peaks translated per condition using pslmap (Materials and methods). *D.melanogaster* peaks are from a previous study from our lab (*Zinzen et al., 2009*). Intersect corresponds to the number of peaks that intersect between both species along with their corresponding percentages according to both species.

• Supplementary file 2: List of 14385 CRMs identified in *D. virilis*. Coordinates correspond to *D. virilis* flybase-R1.2 assembly version (or droVir3 in UCSC)

• Supplementary file 3. List of *D. virilis* CRMs translated to *D. melanogaster*. Listed are the coordinates on *D. melanogaster* (dm3) for the set of *D. virilis* CRMs that were translated using pslmap (Materials and methods). Format is in bed12. For CRMs that map in two locations upon translation, their names were appended by '_x', where x increases from 1 to the number of split events.

• Supplementary file 4. Set of 812 highly conserved CRMs (Multi-ChIP-peak CRMs; Jaccard >= 0.5). Each row correspond to a pair of CRMs with *D. melanogaster* coordinates and ID followed by the corresponding orthologous *D. virilis* coordinates and ID. Coordinates corresponds to dm3 in both cases (as translated *D. virilis* CRMs are compared to their orthologous *D. melanogaster*)

• Supplementary file 5. Same as *Supplementary file 4* for the set of 1130 CRMs with intermediate conservation (Multi-ChIP-peak CRMs; 0 < Jaccard < 0.5).

• Supplementary file 6. Same as *Supplementary file 4* for the set of 500 CRMs with no conservation (Multi-ChIP-peak CRMs; Jaccard = 0).

• Supplementary file 7. List of CAD enhancers used for functional analysis. Listed are in BED format the set of CAD enhancers (Materials and methods) classified as active in the mesoderm (CAD meso) or in any other tissue excluding the mesoderm (CAD Non-meso).

• Supplementary file 8. Evaluation of trained STAP models on 28 TF-ChIP data sets. Pearson correlation coefficient (PCC) between ChIP scores and STAP scores for each TF- and stage-specific model reported.

• Supplementary file 9. Correlations between ΔChIP and ΔSTAP for orthologous CRM pairs and for each corresponding TF and condition.

• Supplementary file 10. Extended description of the 7 orthologous CRMs tested in-vivo for activity in the mesoderm.

• Supplementary file 11. List of primers used to clone *D. melanogaster* and *D. virilis* CRMs.

• Supplementary file 12. List of qPCR primers used for ChIP-qPCR enrichment analysis on *D. virilis* embryos.

## Major datasets

The following dataset was generated:

| Author(s) | Year | Dataset title | Dataset URL | Database, license, and accessibility information |
|---|---|---|---|---|
| Girardot C | 2016 | Virilis chromatin Immunoprecipitation on Drosophila melanogaster embryos during embryogenesis | http://www.ebi.ac.uk/arrayexpress/experiments/E-MTAB-3798/ | Publicly available at the EBI European Nucleotide Archive (accession no: E-MTAB-3798) |

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
