## [Decision Letter]

[Editors’ note: a previous version of this study was rejected after peer review, but the authors submitted for reconsideration. The first decision letter after peer review is shown below.]

Thank you for submitting your work entitled "Evolutionary changes in transcription factor occupancy and its impact on orthologous developmental enhancers' activity" for consideration by *eLife*. Your article has been reviewed by three peer reviewers, and the evaluation has been overseen by a Reviewing Editor and a Senior Editor.

Our decision has been reached after extensive discussion among the reviewers and editors. Based on these discussions and the individual reviews below, we regret to inform you that your work (at least in its current form) will not be considered further for publication in *eLife*. We would, however, consider a resubmission that addresses the concerns raised by the reviewers. There was a consensus that the data collected were interesting and had the potential to move the field forward, but also that the specific analyses of these data and concluding points made in the manuscript were not sufficiently novel to justify publication in this journal. Many references were identified that we felt should have been cited and it is in part with these references in mind that the novelty of the submitted manuscript was lowered (see below). We also discussed how this work could be modified to increase impact and one well-received suggestion was that focusing more on comparing the developmental time courses might provide new insights. For example, can you identify novel functionality based on divergence of TF clusters? If not, then why? We envision that such a revision would likely involve novel analyses that more carefully connect the regulatory networks of matched stages between the species, which was largely untapped as an analysis strategy. Other ideas for reframing this work are suggested by individual reviewers below.

To more accurately discuss how your TF binding analyses confirms TF interaction concepts that have been discussed for some time, see the following recent high impact research studies and reviews:

doi: 10.1016/j.cell.2013.07.007.

doi: 10.1016/j.devcel.2011.09.008.

http://dx.doi.org/10.1016/j.tibs.2014.07.002

These two reviews cite many recent publications that analyse at TF interactions and co-binding in multiple eukaryotes, a selection of which should be smoothly incorporated into a new Discussion, as should the 2013 Cell paper.

In summary, we felt that the data underlying this work could be used to write a paper more suitable for *eLife* in terms of impact, but that performing new analyses and substantially rewriting the manuscript would take more than the two month revision window typically expected by *eLife*. The revised manuscript would also need to be subjected to a full review, thus we decided to reject it in its current form and encourage a resubmission.

Reviewer #1:

Khoueiry et al. provide ChIP-seq in vivo DNA binding of five transcription factors in *D. virilis* embryos at several stages of embryogenesis. They compare these data to existing ChIP-chip data of the same TFs at equivalent stages in *D. melanogaster*. As expected many regions bound strongly in both species are identified, including a set of ~3,400 likely cis regulatory modules (CRMs). The temporal patterns of DNA binding are well conserved over a large evolutionary distance. The most interesting results are the analysis of changes in DNA binding and the presence of recognition motifs between species. Consistent with earlier work, there is a clear correspondence between loss of recognition site motifs and loss of measured DNA binding in vivo for a subset of regions. For a subset of pair-wise combinations of transcription factors, there appear to be modest preferences for DNA binding to co-evolve. Unlike many prior studies, changes in the transcriptional output of enhancers is also examined, using transgenic analysis in *D. melanogaster* of D. virils CRMs.

The work is generally well done and controlled. The broad directions of the conclusions appear reasonable, but I find there to be significant over interpretation. There are several issues that I hope the Authors will address.

1) I am unclear how unique the conclusions are. The "Concluding Remarks" section is unusual. It is two pages in length. It makes sweeping declarations about what the authors believe to be the case, without indicating which claims are novel and which consistent with prior work. It references only one paper for one small point. Many of the claims appear to be consistent with earlier work. For example, it is well known that "the loss or gain of TF binding at an enhancer.… can have diverse effects" and that there is "plasticity of enhancers to evolutionary change". I suggest converting the Concluding Remarks to a proper Discussion section that sets the work in context indicating what is novel and what results support prior research. The Introduction of the manuscript in fact discusses the state of the field quite well.

The authors claim that "cooperative interactions allow a TF to occupy an enhancer despite a loss of its functional motifs". I don't recall that claim being drawn out in the Results. I don't see any evidence for it. As has been repeatedly observed, one often sees TF occupancy in vivo to genomic regions where there is no match to a motif at a given cutoff. But the presence of a lower affinity match in the bound genomic region is not ruled out and indeed is likely. What are the authors basing their claim on and how many instances is there evidence for in the data? How does their result differ from other work?

2) The analysis in Figure 5 is interesting. However, the authors may have over interpreted some of the differences between the more marginally significant and non-significant cases. The exact significance threshold will in part depend on a number of issues unrelated to biology, including the technical quality of the experimental data and the details of the statistical methods used. For example, the authors use Benjamini-Hochberg corrected p-values in Figure 5, which is a less stringent correction than say a Bonferroni correction. Had the Bonferroni correction been used fewer significant co-evolving pairs of TFs would have been identified. The unidirectionality of interactions is open to debate, for example. I would recommend a more measured discussion of the trends, with a focus on the clearest results.

3) An important advantage of this paper is that expression patterns driven by enhancers are compared. However, only seven CRMs are examined, which should temper the somewhat sweeping conclusions made. How were the seven chosen exactly? Were they selected at random from a pool of all CRMs with moderate or poor binding conservation? Or were additional ad hoc criteria used? The authors stress the high conservation in the patterns of output expression between *D. mel*. and *D. vir.*, which is true for some CRMs, but for others the differences in expression appear large, a point that is glossed over and is not captured in the simple summary table.

Reviewer #2:

There are many challenges in understanding when and where genes are expressed in development; identifying all the transcription factors required for organogenesis and their binding specificities, determining whether they bind DNA in cis or function through protein;protein interactions. One strategy is to use evolution and experimental data to decipher a single gene regulatory network. In this manuscript the authors investigate orthologous putative cis-regulatory regions for function using five mesodermal embryonically expressed transcription factors TWI, TIN, *MEF2*, BAP and BIN. Using Chromatin Immunoprecipitation (ChIP) and two distantly related *Drosophila* species – *melanogaster* and *virilis* they identified almost 3,000 conserved predicted CRMs. For seven pairs of predicted enhancers they generated transgenes using the *D. melanogaster* and *D. virilis* predicted CRM and tested activity in *D. melanogaster*. They found that four of the predicted CRM pairs regulate expression concordantly with the SVM predictions (for spatio-temporal activity from the TF binding profile), one pair has partial concordance with the SVM prediction, one pair showed mesodermal expression albeit not the same and the final pair had no expression. Although they have only validated a fraction of the predicted regulatory pairs the paper illustrates a very productive strategy to address questions of the control of gene expression and deserves to be published in *eLife* after the authors consider the following comments.

1) The *D. melanogaster* genome assembly was updated in 2015 from Release 5 to Release 6. They may want to use the latest genomic assembly.

2) The authors use the term Cis-regulatory modules (CRMs) to describe the ChIP bound regions. These should be called predicted or putative CRMs. In the field, a region is designated as a CRMs when it is shown to function to regulate gene expression or transcription rates.

3) The authors used *D. melanogaster* ChIP-chip and compared that to *D. virilis* ChIP-seq. They should discuss any issues of resolution obtained with the two protocols.

4) Regions where all the TFs bind have been called "hot spots". For the regions bound by all five TFs are they also hot spots? Or are they important for mesoderm development?

5) The methods section describing the alignments is quite cryptic. It should be expanded to describe exactly what how they mapped the *D. virilis* peaks onto the *D. melanogaster* sequence. What is the percent similarity of the regions? What cutoffs were used? Are the regions syntenic?

6) Subsection “Conservation of TF binding is factor-dependent”. Need to define "14 conditions". Sounds like all 5 TFs were tested under 14 conditions for a total of 70 experiments. When in fact it is 14 experiments total for the 5 TFs at the developmental stages where they are most active. Please clarify.

7) Could get estimates of how many of the *D. melanogaster* predicted CRMs are functional by comparing to the literature curated experimentally proven CRMs from RedFly and from the Kvon et al., study Nature.

8) It is difficult to determine when the authors are referring to the gene or the protein (e.g. Subsection “Conservation of TF binding is factor-dependent”). Some conventions are to italicize genes and capitalize proteins. So "Twist directly regulates Tinman" would be "TWI directly regulates Tin”. The first time the TF is mentioned the abbreviation should follow in parenthesis. It will make the manuscript more clear.

9) How were the proteins generated for antibody production? Could you expand the Methods section (subsection “ChIP-seq on *D. virilis* embryos”)?

10) Figure 2 and all subsequent gene maps for the *D. virilis* panel shows *D. melanogaster* gene annotations that are poorly drawn – are these gene spans? Why aren't you showing the *D. virilis* gene model annotations (e.g. CG9253 is GJ16136)?

11) Figure 2 and all subsequent gene maps should show genome arm and coordinates (*D. mel)* and scaffolds (*D. vir)* since the scale bar is not clear. Is it only for the *D. virilis* panel or both?

Reviewer #3:

This paper reports the collection and analysis of binding data for 5 key mesodermal TFs at 5 developmental timepoints in *Drosophila virilis*. This dataset complements a comparable dataset collected by the same group in D. *melanogaster* and allows them to conduct systematic analyses of conservation of context dependent TF binding over developmental and evolutionary time. The dataset was carefully collected and will be useful to the fly community, and likely useful to the broader computational community studying enhancer evolution.

The paper is framed as a general study on enhancer evolution, but in my opinion, the most interesting and novel contribution is a direct test of the "TF collective" model that the authors put forth previously. In this model, TFs can bind to DNA via contacts with other TFs, such that a cognate DNA recognition motif is not strictly required. A clear prediction of this model is that over evolutionary time, binding of such a TF may be conserved while the underlying motif is not. They indeed observe this behavior for Tinman, and by tracking co-binding and loss of co-binding, identify the TFs that may work collectively with Tinman. I believe the article would be more effective if crafted more directly around testing this model, with the additional observations about enhancer evolution framed as observations that emerge from the dataset and provide additional evidence for well-established concepts in the field.

As currently written, the paper marches through many well-established concepts in the field and the novelty of the work is a bit lost. For example, their functional tests of 7 pairs of orthologs (14 reporter constructs total) demonstrate the concept that we cannot yet predict the relationship between enhancer sequence, TF occupancy and function. I think this is a key puzzle for the field, so more evidence of interesting discrepancies is welcome, but it is not a major take home message of the paper, as put forth in the abstract.

My only major technical criticism concerns their identification of orthologous enhancers based (apparently) just on synteny. They identify ChIP peaks separately in each species and then use pslMap to find corresponding peaks between *D. mel* and D.vir, which they assign as orthologous and use for all of the subsequent analyses. Given that enhancers with similar activity can move genomic location (e.g. Kalay and Wittkopp, PLoS Genetics 2010), assigning orthology based on synteny may not be straightforward. This is a critical part of their study, and they should clearly justify their strategy, and its potential pitfalls and impacts on their downstream analyses. It may be useful to cross-check their assignments by searching locally for better matches to *D. mel* enhancers based on sequence or occupancy profile; this would potentially give a sense of the error in their synteny based assignment.

[Editors’ note: what now follows is the decision letter after the authors submitted for further consideration.]

Thank you for submitting your article "Uncoupling evolutionary changes in DNA sequence, transcription factor occupancy and enhancer activity" for consideration by *eLife*. Your article has been reviewed by three peer reviewers (one from the original submission plus two new reviewers), and the evaluation has been overseen by Patricia Wittkopp as the Senior Editor. The following individuals involved in review of your submission have agreed to reveal their identity: David N Arnosti (Reviewer #3).

The reviewers have discussed the reviews with one another and the Reviewing Editor has drafted this decision to help you prepare a revised submission.

Summary:

The changes made to this work have clarified its novel contributions to the field. The revised manuscript effectively summarizes the way in which the authors focus on a core set of putative CRM for further analysis, concisely indicates how overall patterns for binding show a high degree of conservation in developmental windows, and the extent to which DNA sequence is able to predict occupancy on CRMs classified as highly bound or not, and specific cases for which the "collective" model of enhancer occupancy is substantiated (particularly for elements involving Tinman). This last point represents substantial progress over their previous studies, in which the lack of observable cognate binding motifs was suggested to be evidence for occupancy driven largely by non-DNA mediated interactions with neighboring factors. Here, the authors clearly show that for this group of CRM, there are a sizeable number that appear to fit this category, and they have sufficient information to assess the frequency with which altered binding of factor A is linked to changes in motifs for factor B. We also appreciated the more detailed and measured statistical analysis of Figure 5 and that the approach for choosing CRMs for transgenic analysis was made more transparent and the results more thoroughly discussed. Overall, the revisions make this paper a strong contribution to the literature that will be of great interest to the evo/devo and genomic research communities. We do, however, also have some additional questions/concerns that need to be addressed prior to publication.

Essential revisions:

1) The use of FDR is standard for the field and must be provided in addition to any other metrics the authors wish to present. We do not anticipate this will change the results. In addition, if the IDR results remain, please add language recognizing that it is a relative measure of reproducibility between treatment replicas and does not control for systematic biases due to, for example, preferential amplification of some genomic regions over others.

Additional details including the control used and threshold applied must also be provided for readers to properly interpret the q value calculations presented.

2) The authors should make clear that only replica antiserum against the same set of epitopes was used for each protein. In contrast, studies by other groups have used two sera reacting with different portions of the ChIP'd protein as a "gold standard" for ChIP-seq. The revised manuscript must clearly state what might be missed by using only serum to common epitopes, rather than two antisera recognizing different parts of the same protein. We encourage the authors to use reagents recognizing different epitopes or domains of the proteins of interest in the future, as other researchers have done.

3) Additional methodological detail is needed. The Materials and methods are missing some important information; details on chromatin preparation (presumably similar or identical to earlier work, but not cited), numbers of reads obtained from different ChIP-seq experiments.

4) It seems the authors are considering only direct cooperativity between proteins. Indirect cooperativity is also a major mechanism where the binding of one protein impacts the binding of another protein via an effect of nucleosome occupancy and should be mentioned

5) The claim that *Mef2* binds a less conserved set of peaks in early time points is somewhat driven by the large number of peaks in *D. melanogaster* at time point 2, compared to *D. virilis*, which is in contrast the rest of the data, where *D. virilis* peaks far outnumber *D. melanogaster* peaks. Is this difference because of differential expression dynamics/levels of *Mef2* in the two species? In general, do we know if the expression dynamics/levels of these factors are conserved between these species? How might this impact the measured conservation of binding?

6) Please provide additional pieces of information to confirm the claims that Tin binding is less dependent on sequence as binding complexity increases. For example, no statistical test was provided for the motif density comparison in subsection “TF binding site divergence is tolerated in the context of collective TF binding”. Please provide the reader with more information as to how to compare the numbers of red/green dots in Figure 4; there seems to be different numbers of data points in the two plots and fractions are not reported/compared.

7) It would be interesting to know if the SVM from Zinzen 2009 already accounts for the fact that differential TF occupancy can lead to the same expression output in the manner observed here. For example, if the SVM were given the binding profile of the D. vir CRM 10323, would it predict SM expression? Or given the D. vir CRM 2469 binding profile, would it predict VM expression? This would indicate whether this divergence in TF binding leading to the same outcomes is already observed within *D. melanogaster* or whether new and unique regulatory logic is being employed in *D. virilis*, which may indicate the involvement of other TFs.

---

## [Author Response]

[Editors’ note: the author responses to the first round of peer review follow.]

*Reviewer #1:*

*Khoueiry et al. provide ChIP-seq in vivo DNA binding of five transcription factors in D. virilis embryos at several stages of embryogenesis. They compare these data to existing ChIP-chip data of the same TFs at equivalent stages in D. melanogaster. As expected many regions bound strongly in both species are identified, including a set of ~3,400 likely cis regulatory modules (CRMs). The temporal patterns of DNA binding are well conserved over a large evolutionary distance. The most interesting results are the analysis of changes in DNA binding and the presence of recognition motifs between species. Consistent with earlier work, there is a clear correspondence between loss of recognition site motifs and loss of measured DNA binding in vivo for a subset of regions. For a subset of pair-wise combinations of transcription factors, there appear to be modest preferences for DNA binding to co-evolve. Unlike many prior studies, changes in the transcriptional output of enhancers is also examined, using transgenic analysis in D. melanogaster of D. virils CRMs.*

*The work is generally well done and controlled. The broad directions of the conclusions appear reasonable, but I find there to be significant over interpretation. There are several issues that I hope the Authors will address.*

*1) I am unclear how unique the conclusions are. The "Concluding Remarks" section is unusual. It is two pages in length. It makes sweeping declarations about what the authors believe to be the case, without indicating which claims are novel and which consistent with prior work. It references only one paper for one small point. Many of the claims appear to be consistent with earlier work. For example, it is well known that "the loss or gain of TF binding at an enhancer can have diverse effects" and that there is "plasticity of enhancers to evolutionary change". I suggest converting the Concluding Remarks to a proper Discussion section that sets the work in context indicating what is novel and what results support prior research. The Introduction of the manuscript in fact discusses the state of the field quite well.*

We wrote the paper in the style of a mixed ‘results and discussion’ section, followed by a short concluding remarks, as the paper is quite long and we had also described the state of the field in the introduction, which we appreciate the reviewer’s comment on. Although those are the main reasons, it also stems from a personal preference of not putting too much weight on the Discussion sections of the published literature. However, given the reviewers comment, we have now changed the format to a discussion.

*The authors claim that "cooperative interactions allow a TF to occupy an enhancer despite a loss of its functional motifs". I don't recall that claim being drawn out in the Results. I don't see any evidence for it. As has been repeatedly observed, one often sees TF occupancy* in vivo *to genomic regions where there is no match to a motif at a given cutoff. But the presence of a lower affinity match in the bound genomic region is not ruled out and indeed is likely. What are the authors basing their claim on and how many instances is there evidence for in the data? How does their result differ from other work?*

We apologize for the lack of clarity in our statement leading to this confusion. The statement is suggested by a combination of two main observations, described below. But first, just to clarify, to avoid exactly this motif cut-off issue raised by the reviewer, we purposely used STAP, a threshold free approach. Our observations are therefore based on first estimating the net motif presence in a CRM using STAP, which utilizes a thermodynamics-based model of occupancy taking both high and low affinity sites into account. These estimates are then compared between orthologs and this ‘ΔSTAP’ score is then used as a measure of motif loss or conservation.

Our two main observations are the following: First, Figure 4 shows that changes in Tin binding (ΔChIP) between orthologous CRMs is relatively well explained by changes in Tin motif presence (ΔSTAP) when we consider ‘singleton’ CRMs, but that this relationship is significantly weaker when examining CRMs occupied by multiple TFs (‘high occupancy’ class). We interpret this as implying that Tin binding is less dependent on its motif in the context of other functionally related ‘partner’ TFs due to protein::protein interactions, consistent with the TF collective model. We note that this is not a simple matter of inadequate motif modelling, since the same score of motif presence is used in examining the two classes of CRMs. Moreover, the observation is not merely about TF binding and its relation to motif presence but about change in TF binding and its relation to change in motif presence. We have observed a statistically significant *difference* in how this expected relationship manifests in singleton CRMs versus high occupancy CRMs.

Second, motivated by the above observation and its possible implications, we tested if change in Tin binding (ΔChIP) between orthologous CRMs is significantly associated with changes in motif presence of a secondary mesodermal TF such as Twi, *Mef2*, Bap or Bin, all of which are known to be important regulators of the system under study. As shown in Figure 5, we found that this is indeed the case, for Bap (uncorrected p-value 2.09E-4), Bin (uncorrected p-value 1.68E-2), and marginally so for Twi (uncorrected p-value 4.4E-2).

Taken together, these finding suggest that conservation of TF occupancy, i.e. whether a TF’s ChIP peak remains occupied in an orthologous CRM, is partly determined by whether a secondary TF’s motif is conserved or not. We have now reworded the quoted statement in the above comment to reflect this interpretation more accurately. We have also included a model figure in the discussion (Figure 7), which highlights what has been observed in other studies and what is new in ours. It shows three different motif:TF binding relationships, one of which (TF collective) is novel to this study.

We believe the novelty of our claim is in its evolutionary context, in a regulatory system where we know that the TFs examined are genetically required for this tissues development, and therefore these TF bound enhancers are very likely to be functionally relevant to the gene’s expression in this spatio-temporal domain. The reviewer correctly points out that the role of secondary TFs in shaping occupancy of a TF has been amply recorded in the literature. So the evolutionary loss of one TF’s binding can be due to the loss of another TFs binding, while the first TF’s binding site is conserved. A number of groups have observed this (which we reference), and we see this as well in our data (now highlighted in the new Figure 4, and Figure 7). But, importantly, we also find examples of the opposite: conservation of a TFs occupancy when its motif has diverged. This is consistent with the increased flexibility in TF occupancy predicted by the ‘TF collective model’, and this the first large scale cross-species study showing that that is the case (Figure 4 and Figure 7). The TF collective model implies a greater neutral neighbourhood for evolutionary changes than has been noted. We have expanded on this aspect of the paper as suggested by reviewer 3.

*2) The analysis in Figure 5 is interesting. However, the authors may have over interpreted some of the differences between the more marginally significant and non-significant cases. The exact significance threshold will in part depend on a number of issues unrelated to biology, including the technical quality of the experimental data and the details of the statistical methods used. For example, the authors use Benjamini-Hochberg corrected p-values in Figure 5, which is a less stringent correction than say a Bonferroni correction. Had the Bonferroni correction been used fewer significant co-evolving pairs of TFs would have been identified.*

We fully agree with the reviewer’s point. Selecting significant thresholds and/or the multiple testing correction method is subject to both common practices (e.g. selecting a 0.05 significance threshold) and the analysis context (FDR thresholds up to 20% are often used in screening contexts when eliminating false positives is less important than missing true positives). Although the Benjamini-Hochberg procedure is a less stringent correction than the Bonferroni correction, the overall conclusions are not strongly affected by using the Bonferroni correction (see table below). Only the Bin-Bap associations become not significant with the Bonferroni correction (0.04 BH corrected p-value vs 0.16 Bonferroni corrected p-value). Given the low number of counts, we decided to use the well-established Benjamini-Hochberg correction, which evaluates the False Discovery Rate (as opposed to controlling the Family Wise Error Rate in the Bonferroni case) method. We fully agree that the significance of the Bin-Bap association should be regarded with caution. We now mention this association, but do not expand on it.

A comparison of the Benjamini-Hochberg (BH) and Bonferroni corrected p-values is provided for all pair-wise comparisons below (cons = conserved; Exp = expected). Yellow indicates the 4 significant cases under discussion.

TF1TF2TF1 & TF2 consTF1 cons TF2 lostTF1 lost TF2 consTF1 & TF2 lostExp TF1 & TF2 lostPvalBH (FDR) Bonfer ronitwibin1171042822237.55E-017.55E-01 1.00E+00twitin443854020112.12E-037.06E-03twibap6496929246.45E-021.29E-01 6.45E-01 twimef24202976129361.10E-011.83E-01 1.00E+00bintin104326030251.28E-011.83E-01 1.00E+00binbap49671241341.58E-023.95E-021.58E-01binmef21575912656535.06E-015.62E-01 1.00E+00tinbap9082147286.00E-126.00E-116.00E-11tinmef23231546769491.15E-045.74E-041.15E-03bapmef2671711743392.77E-013.46E-01 1.00E+00

*The unidirectionality of interactions is open to debate, for example. I would recommend a more measured discussion of the trends, with a focus on the clearest results.*

This is a fair point, and we can now see how the table in Figure 5 (now Figure 5) was confusing. We previously reported all interactions, while we were primarily interested in testing the associations between the binding loss of TF1 and the loss of sequence support for a TF2, in particular for the Tin-Twist, Tin-*Mef2* and Tin-Bap pairs (and to a lesser extend Bin-Bap) i.e. 6 tests (or 8 tests when considering Bin-Bap), but we presented the results of all 20 directional tests. We acknowledge the reviewer’s advice to focus on the clearest results- we now realize it was detracting from our main point to present the results from all 20 tests and we have therefore modified the table to present the clearest results, which is for Tin-Twist, Tin-*Mef2* and Tin-Bap pairs only. In this context, the p-values for *Mef2*-Tin (1.96E-4), Tin-Bap (2.09E-4) and Bap-Tin (8.44E-4) remain clearly significant even under Bonferroni correction, while the p-values for Twi-Tin (4.3E-2) and Tin-Twi (4.4E-2) should be considered with care. The manuscript was already cautious about the Tin-Twi results (“Our results also suggest …”) and we therefore left this part of the text unmodified.

*3) An important advantage of this paper is that expression patterns driven by enhancers are compared. However, only seven CRMs are examined, which should temper the somewhat sweeping conclusions made. How were the seven chosen exactly? Were they selected at random from a pool of all CRMs with moderate or poor binding conservation? Or were additional ad hoc criteria used?*

Thank you for recognizing the importance of testing the expression patterns of the orthologous enhancers. In our manuscript we tried to move away from simply examining conserved or non–conserved binding to rather compare what effect conservation/divergence in TF binding has on enhancer activity – which is in the end is what selection will be acting on. We note that ‘only seven CRMs’ means in effect generating 14 transgenic strains, as we tested the enhancer from each species.

The CRM pairs chosen for in vivo testing were not chosen randomly, as we specifically wanted to test the assumption that divergence of occupancy means diverged enhancer activity. We state the criteria used to select the enhancers for testing in the manuscript (subsection “Linking evolutionary changes in TF occupancy to their effects on enhancer activity”): “(1) high SVM specificity (spec. > 0.98) of the predicted spatio-temporal expression of the *D. melanogaster* CRM, (2) medium to low Jaccard index (J <= 0.5) indicative of a medium to high turn-over in TFs binding on orthologous CRMs (Table S10) and (3) good alignment of orthologous loci at the corresponding CRM regions” to ensure that the true orthologous enhancer is cloned. We only retained pairs bound in at least 5 different conditions (altogether) to avoid reviewing CRM pairs with marginal changes in their binding profiles. Of the 161 orthologous pairs passing all these criteria, we then checked manually for the presence of a good alignment of orthologous loci using the Integrative Genome Viewer (IGV). These “objective” qualitative criteria ensured that the integrity of the CRM and its surrounding locus is preserved and constitute thus a valid option for the subsequent *in-vivo* analysis. We now also add these criteria to the text.

*The authors stress the high conservation in the patterns of output expression between D. mel. and D. vir., which is true for some CRMs, but for others the differences in expression appear large, a point that is glossed over and is not captured in the simple summary table.*

Yes, for some enhancers the expression is highly conserved, while for others it has partially diverged, having conserved activity in some tissues and non-conserved in others. We have now extended the text to highlight this. For some enhancer pairs this may be due to the binding of other TFs, which were not assayed in our ChIP-seq, on the tested CRM pairs. The text now states “We note that, although the TF binding is highly conserved, three pairs (*D. mel* CRM-6087/*D. vir* CRM-459; *D. mel* CRM-4682/*D. vir* CRM-12291; *D. mel* CRM-4725/*D. vir* CRM-12156) showed only a partial overlap in their activity, probably due to the binding of additional TFs not assessed here.”

*Reviewer #2:*

*[…] 1) The D. melanogaster genome assembly was updated in 2015 from Release 5 to Release 6. They may want to use the latest genomic assembly.*

While we agree that this would be great (and the lab is migrating to Dm6), we regret that this is not possible for this project. The dm6 version was released in 2015, as the reviewer mentioned, when a substantial amount of the analysis was already completed, including all reads alignment, motif discovery analysis, chains used for peak translation between species as well as many other complementary analysis. However, we will provide Dm6 peaks, generated by liftover to dm6 coordinates as it is generally done, as well as Dm3 peaks on our web page when published, as we routinely do for all studies.

*2) The authors use the term Cis-regulatory modules (CRMs) to describe the ChIP bound regions. These should be called predicted or putative CRMs. In the field a region is designated as a CRMs when it is shown to function to regulate gene expression or transcription rates.*

We completely agree with the reviewer. We typically use ChIP-CRM to describe putative enhancers, and enhancers when the region has been shown to function as an enhancer in transgenic embryos. For this data, we first defined this for the *melanogaster* data in 2009 “…8,008 ChIP defined *cis*-regulatory modules (ChIP-CRMs), henceforth called CRMs”. In the current paper we state “…8,008 ChIP defined putative CRMs (ChIP-CRMs), henceforth called CRMs”. We state this “CRM” definition at the outset in the paper (subsection “Conserved enhancer occupancy is associated with enhancer function”).

*3) The authors used D. melanogaster ChIP-chip and compared that to D. virilis ChIP-seq. They should discuss any issues of resolution obtained with the two protocols.*

This was something that we were concerned with. We show in Figure—figure supplement 1that signals of TF occupancy identified by ChIP-chip globally correlates with peaks identified by ChIP-seq for the same factor (*Mef2*) at the same time point (6-8h) and in the same species (*D. melanogaster*). This indicates that the overall resolution difference between the two technologies does not have a major effect. To control for the different base-pair resolution of ChIP-seq compared to ChIPchip, we performed the bulk of our analyses based on the ChIP peak center, extended by a defined number of bases, to minimize any resolution effects. For instance, CRM calling in both species was based on TF peak centers and defining CRMs upon clustering nearby cases (Methods). Also, motif discovery considers peak centers extended by 100bp.

*4) Regions where all the TFs bind have been called "hot spots". For the regions bound by all five TFs are they also hot spots? Or are they important for mesoderm development?*

Kvon et al., nicely demonstrated that HOT regions function as patterned developmental enhancers in their Genes Dev. 2012 paper. In addition, in our Zinzen et al., 2009 paper, one of the CRMs we demonstrated functions as an enhancer in transgenic is a CRM bound by the 5 different TFs studied here (Figure 5, CRM 4726). Therefore, both HOT regions and regions bound by our 5 mesodermal factors function as enhancers, and are likely important for development. To specifically answer the reviewer’s question, we split the HOT regions (downloaded from http://www.modencode.org/publications/files/fly/DataS8.gff) and, as described in Kvon et al., (Genes Dev. 2012), split them into HOT (complexity score strictly >8, 1854 regions), WARM (3 < complexity score ≤8 and strictly, 8467 regions) and COLD (complexity score ≤3, 28241 regions) groups. We then assessed if the 137 Dmel ChIP-CRMs bound by the 5 TFs overlaps more with the HOT group compared to the other two groups. Requesting for at least 50% overlap of the CRMs, we find that 50, 48 and 20 CRMs bound by the 5 TFs overlap with HOT, WARM and COLD regions, respectively. These numbers indicate that Dmel ChIP-CRMs bound by the 5 TFs are equally associated with HOT and WARM regions, confirming the results from Kvon et al., but also that 87 of the 137 CRMs (63%) are not HOT regions.

*5) The methods section describing the alignments is quite cryptic. It should be expanded to describe exactly what how they mapped the D. virilis peaks onto the D. melanogaster sequence. What is the percent similarity of the regions? What cutoffs were used? Are the regions syntenic?*

We have now expanded this part of the Methods section. The regions are a bit more than syntenic, as captured by the alignment. We did not use a cutoff here. We have elaborated and described the pslmap algorithm in the methods. Briefly, Pslmap uses a precomputed alignment, as the liftover used in He, Bardet et al., and assigns orthologous information based on alignment scores. It does this base-by-base. The advantage over liftover (which does block-level mappings) is that pslmap takes into consideration rearrangements including insertions and deletions, allowing expansion or contraction of block sizes.

*6) Subsection “Conservation of TF binding is factor-dependent”. Need to define "14 conditions". Sounds like all 5 TFs were tested under 14 conditions for a total of 70 experiments. When in fact it is 14 experiments total for the 5 TFs at the developmental stages where they are most active. Please clarify.*

We agree, we have now rephrased it as follows “The number of time-points examined for each TF reflects the developmental stages that the TF is expressed, totaling 14 conditions (TF and time point).

*7) Could get estimates of how many of the D. melanogaster predicted CRMs are functional by comparing to the literature curated experimentally proven CRMs from RedFly and from the Kvon et al., study Nature.*

We agree – we actually did such a comparison in Figure 2, comparing experimentally validated and curated CRMs from RedFly and Kvon et al., study (Figure 2). We state, in methods subsection “Enrichments of characterized enhancers”: To assess if our conserved set of CRMs are enriched for known mesodermal enhancers, we extended our previously built CRM Activity Database (CAD (Zinzen, 2009) by adding the activity of active tiles from Kvon and colleagues (Kvon, 2014) and a set of new entries from RedFly (Gallo, 2011).

*8) It is difficult to determine when the authors are referring to the gene or the protein (e.g. Subsection “Conservation of TF binding is factor-dependent”). Some conventions are to italicize genes and capitalize proteins. So "Twist directly regulates Tinman" would be "TWI directly regulates Tin”. The first time the TF is mentioned the abbreviation should follow in parenthesis. It will make the manuscript more clear.*

We now updated the text to reflect the reviewer’s valid point. All protein instances are now capitalized and abbreviated (i.e. Twi for Twist) and gene instances are lower case and italicized (i.e. *twi* for the twist gene). In this case Twi (the protein) regulates *tin* (the gene).

*9) How were the proteins generated for antibody production? Could you expand the Methods section (subsection “ChIP-seq on D. virilis embryos”)?*

The antibodies directed against the *virilis* proteins were generated as follows – first the protein was expressed in, and purified from, *E. coli* as a SUMO fusion protein. This was then injected into rabbits, in the EMBL animal house, to raise antibodies. We then used the SUMO tag purified protein to purify the antibodies. We now add more information on this to the method section.

*10) Figure 2 and all subsequent gene maps for the D. virilis panel shows D. melanogaster gene annotations that are poorly drawn – are these gene spans? Why aren't you showing the D. virilis gene model annotations (e.g. CG9253 is GJ16136)?*

Yes, those are gene spans. The orthologous information we had was mapping *D. mel* genes to *D. vir* gene spans. We updated the figures now to include all gene information with exons and introns clearly visible.

*11) Figure 2 and all subsequent gene maps should show genome arm and coordinates (D. mel) and scaffolds (D. vir) since the scale bar is not clear. Is it only for the D. virilis panel or both?*

The same scale was used for screenshots of both species. We now state that in figure legends by saying (i.e. in Figure 2 legend): “The same scale was used for both species and indicated on the lower right corner”.

*Reviewer #3:*

*[…] My only major technical criticism concerns their identification of orthologous enhancers based (apparently) just on synteny. They identify ChIP peaks separately in each species and then use pslMap to find corresponding peaks between D. mel and D.vir, which they assign as orthologous and use for all of the subsequent analyses. Given that enhancers with similar activity can move genomic location (e.g. Kalay and Wittkopp, PLoS Genetics 2010), assigning orthology based on synteny may not be straightforward. This is a critical part of their study, and they should clearly justify their strategy, and its potential pitfalls and impacts on their downstream analyses. It may be useful to cross-check their assignments by searching locally for better matches to D. mel enhancers based on sequence or occupancy profile; this would potentially give a sense of the error in their synteny based assignment.*

We also shared the reviewer’s concern in how best to assign orthologous enhancers, and actually explored different ways to do this. “pslmap”, unlike liftover used previously (i.e. He, Bardet et al), does a “base-by-base mapping, which may insert or delete bases within a block”. This is thus not a “simple synteny” approach, where an anchor point is used to define syntenic elements. The mappings produced by liftOver is block-level, which may expand or contract the size of blocks and is thus not ideal for defining orthologous regions between distantly related species.

Additionally, and most importantly, we translated CRMs from *D. virilis* to *D. melanogaster* and not peaks or reads. This is important, as if one translates the reads first (from D. vir to *D. mel)* and then call peaks, the peak calling is significantly affected by the liftover of each short read in such distant species. Therefore, we defined significant TF peaks in *D. virilis*, and defined *D. virilis* CRMs, after lifting over the *D. virilis* CRMs to *D. mel* we maintain the binding information (i.e. number and quality TFs bound) to that of *D. virilis*, as measured. We are not assigning TF peak to peak (*D. virilis* to *D. melanogaster*), which would be the case if we translated the data at the level of peaks. By translating CRMs, for example, a Bap TP3 peak in *D. virilis* (7-10h) is considered conserved if it falls in *D. virilis* CRM whose translation overlaps with a *D. mel* CRM with a Bap TP3 (6-8h) in it, independent of whether both peaks overlap. Figure 2 gives a faithful example of that. For instance, Figure 2 upper panel shows a strong conservation pair with a Jaccard index of 0.66 with both the red and green TFs conserved, although not aligned. This shows that we did not restrict our conservation analysis to aligned peaks (strict synteny) from both species, but rather allow assignments of conserved peaks by “searching locally”, in the CRM element itself. Zooming out, it is true (and we are aware) that enhancers themselves can move around significantly during evolution. An enhancer, for example 5’ of the gene, might move into an intron or 3’, just as an example. In this case, we would not have called an orthologous CRM, as there would be no CRM overlapping our *D. virilis* element. We are aware of this, but our goal here was to obtain a stringent set of orthologous elements (high specificity), rather than casting the net (genomic distance) wider with the risk of defining artificial orthologous pairs.

Given the reviewers suggestion we have now examined this issue, i.e. how much ‘CRM movement’ is there locally, by assessing the frequency of binding events in CRMs in close proximity to our orthologous CRM pairs with conserved or lost binding. If a locally better match CRM exists, in terms of matching patterns of TF occupancy, the average distance to those matches should be small, taking the compact genome of both assessed *Drosophila* species. For this, we split CRMs assigned as orthologous into two categories: Those with a conserved binding for the assessed condition (TF and Time point) and those with a binding loss in *D. mel* or in *D. virilis* (they are still defined as CRMs since they have a binding for another factor). In each case, we calculated the distance to the closest binding event (i.e. for Twi_TP1, we calculated the distance to the closest Twi binding at the first time point). In all 14 conditions, the median distance to the closest TF exceeded 5kb when we look in *D. melanogaster*. This was also the case when we looked in *D. virilis*, except for Twi at the second time point and Tin at both time points where the medians were between 1kb and 5kb. As a reminder, the median size of *D. virilis* and *D. melanogaster* CRMs is 200bp.

This analysis (Figure 8)shows that, in average, we need to search for additional binding events (a putative orthologous CRM) as far away as 5kb, which as you know is a large distance for *Drosophila*’s relatively compact genome. Importantly, in 68% of these cases we had already assigned the distant CRM to an orthologous CRM pair that overlapped in both species. In these cases we would therefore have to consider two-to-one cross-species matches, one overlapping and one several kb away. For comparison, the median spacing between all consecutive CRMs is 3.2kb in *D. virilis* (Mean = 10.5kb, 14386 CRMs) and 3.3kb in *D. melanogaster* (Mean = 14.4 kb, 8008 CRMs). We therefore feel we risk assigning false pairs at larger distances. Moreover, owing to their modular and separable nature, CRMs as close as 500 bp can have completely distinct activity. For example, we previously showed that the 1.2 kb Combo region (Zinzen et al., 2009) harbors two CRMs, with some shared binding, where one drives activity in the somatic muscle and the other in the visceral muscle. Finally, given that we already have nearly 3000 orthologous pairs using a conservative overlapping approach, we feel this is ample numbers to examine the global properties of orthologous binding events. For other, more ‘loose’ definitions of orthologous enhancer pairs, all data will be available for future studies to explore.

Author response image 1.Boxplots represent the distribution of distances to closest peaks (in logarithmic scale) for each of the 14 conditions assayed.The distribution of distances of a TF binding event lost in *D. mel* (BindingLostInDmel) at the indicated condition (i.e. Twi_TP1) to the closest peak for the identical condition in another CRM. Similarly, BindingLostInDvir shows the distance distribution of a TF binding event lost in *D. virilis* at the indicated condition to the closest peak for the identical condition in another CRM. BindingConsDmelCoor shows distance distribution for *D. melanogaster* CRMs that have their binding conserved and similarly for BindingConsDvirCoor..**DOI:**
http://dx.doi.org/10.7554/eLife.28440.025

In addition to the above analysis, we calculated the absolute number of CRMs that have a close binding event, for different distances and categories. Distances to closest binding event were assigned to one of the 5 categories (<100bp, 100-200bp, 200-500bp, 500-1000bp and > 1000bp). Again, one must go to distances >1kb from the CRM to find region bound by one of the same TFs at the same time point (TP).

Author response image 2.Count of CRMs with a close binding event falling in one of the five distance categories.y-axis shows absolute counts (also displayed as numbers above corresponding bars) and x-axis the 5 distance categories. The analysis was done for all 4 categories of CRMs: Binding lost in *D. mel*, binding lost in *D. virilis*, binding conserved (for *D. mel* CRMs) and binding conserved (for *D. virilis* CRMs).**DOI:**
http://dx.doi.org/10.7554/eLife.28440.026

[Editors' note: the author responses to the re-review follow.]

*Essential revisions:*

*1) The use of FDR is standard for the field and must be provided in addition to any other metrics the authors wish to present. We do not anticipate this will change the results. In addition, if the IDR results remain, please add language recognizing that it is a relative measure of reproducibility between treatment replicas and does not control for systematic biases due to, for example, preferential amplification of some genomic regions over others.*

*Additional details including the control used and threshold applied must also be provided for readers to properly interpret the q value calculations presented.*

We apologize for the lack of peak calling details in the Materials and methods.

We have amended the manuscript to reflect the reviewer’s point as follows:

The main text was changed from “The number of high confidence peaks, called at a 1% Irreproducible Discovery Rate (IDR) threshold (Materials and methods)” to “The number of high confidence peaks, called at a 1% Irreproducible Discovery Rate (IDR, a measure ensuring equivalent reproducibility between replicas) threshold, corresponding to peak calling q-value < 0.001 in all conditions (Materials and methods)”.

Additionally, the method section “ChIP-seq sample processing” was enriched with all peak-calling details.

*2) The authors should make clear that only replica antiserum against the same set of epitopes was used for each protein. In contrast, studies by other groups have used two sera reacting with different portions of the ChIP'd protein as a "gold standard" for ChIP-seq. The revised manuscript must clearly state what might be missed by using only serum to common epitopes, rather than two antisera recognizing different parts of the same protein. We encourage the authors to use reagents recognizing different epitopes or domains of the proteins of interest in the future, as other researchers have done.*

We thank the reviewer for highlighting this important technical issue. In fact, we have used two different sera for two of our transcription factor ChIP, *Mef2* and Twist. In the case of *Mef2*, the sera were generated against the DNA binding domain while for Twist we used the full-length protein. Although we cannot guarantee that the sera from both animals targets different portions of the protein, it is also not a 100% guaranteed that both sera, i.e. that the two different animals will raise antibodies targeting the same portion of the protein as we generated polyclonal antibodies. For the other three transcription factors assayed here, we have only used one antibody, as there is only one good ‘ChIP grade’ antibody available. While we completely agree with the reviewer’s point, that the best practice is to use two antibodies generated from different parts of the protein, we note that the vast majority of ChIP experiments against *Drosophila* TFs have only used one antibody, as there are very few antibodies commercially available. In all of our previous ChIP studies in *melanogaster* we have used two antibodies (although again they were generated against the same protein domains (usually the entire protein)).

To address the reviewers point, we have now added the following to the ChIP Methods: “Two independent biological replicates were generated for each transcription factor at each time-point, using two independent antibodies when possible (for Twi and *Mef2*, targeting the full-length and DNA binding domains, respectively). Although we have no guarantee that both sera target different portions of the protein, these polyclonals were generated in different rabbits and therefore have a different background level of IgG antibodies in the sera.”

*3) Additional methodological detail is needed. The Materials and methods are missing some important information; details on chromatin preparation (presumably similar or identical to earlier work, but not cited), numbers of reads obtained from different ChIP-seq experiments.*

We apologize for the lack of information related to this matter. We have now added this to the materials and method: “Chromatin from *D. virilis* staged embryos was extracted as described previously for *D. melanogaster* (Sandmann et al., 2006).”

Also, we extended Table S1 to include the final number of reads (duplicates and multi-mapped reads removed) for both replicates used for each TF, time point combination as well as for the matching input replicates.

*4) It seems the authors are considering only direct cooperativity between proteins. Indirect cooperativity is also a major mechanism where the binding of one protein impacts the binding of another protein via an effect of nucleosome occupancy and should be mentioned.*

We thank the reviewer for pointing to this additional alternative. You are right, we did mainly focus on direct protein:protein and protein:DNA interactions. To highlight this alternative, we now mention at the end of paragraph “Co-divergence in TF occupancy identifies potential cooperative binding of TFs”, that “…the occupancy of Tin is highly cooperative, which could involve direct cooperativity that is dependent on the interplay between protein: DNA and protein: protein interactions, although we note this could also involve indirect co-operativity through co-factor recruitment or nucleosome displacement”.

*5) The claim that Mef2 binds a less conserved set of peaks in early time points is somewhat driven by the large number of peaks in D. melanogaster at time point 2, compared to D. virilis, which is in contrast the rest of the data, where D. virilis peaks far outnumber D. melanogaster peaks. Is this difference because of differential expression dynamics/levels of Mef2 in the two species? In general, do we know if the expression dynamics/levels of these factors are conserved between these species? How might this impact the measured conservation of binding?*

We agree with the reviewer that the differences at this time-point are striking. We don’t think this is due to a difference in expression dynamics between the two species. We actually spent quite some effort in assessing that, in order to select comparable time-windows for the experiments. Our in-situ (data not shown) and immunostaining data (Figure 1 and Figure—figure supplement 1) don’t show any delay in the expression of *mef2* transcripts or *Mef2* protein. Actually the expression pattern of *Mef2* is beautifully conserved, even though it is complex and very dynamic.

One possibility is that the binding of *Mef2*, rather than its expression, is slightly delayed in *D. virilis* compared to *D. melanogaster*. There is a *D. melanogaster* protein, Him, that was shown to inhibit *Mef2* binding early in development (Liotta et al., 2007) – although the timing of action of Him doesn’t fit this observation, it indicates that *Mef2*’s ability to bind to its targets can be inhibited by other proteins. Perhaps there is another protein or mechanism in *D. virilis*. To account for this option, we have now added the following: “This could represent a shift (or delay) in the initiation of *Mef2* binding between species, rather than in *Mef2* expression (which is conserved as seen in Figure 1 and Figure—figure supplement 1), which would impact the ability of *Mef2* to bind at early stages.”

Although standard in-situ hybridization and immune-stains are not fully quantitative, we found no observable differences in the levels of *Mef2* at equivalent stages between species, although *Mef2* expression does increase over time within each species, as previously reported. One alternative possibility is that *D. melanogaster Mef2* antibody used here on *D. virilis* cross-react less efficiently at earlier time points, where *Mef2* is less “functionally” required, as known from *Mef2* LOF mutants. However, this is less likely to be the case since the same antibody was used efficiently for later time points.

*6) Please provide additional pieces of information to confirm the claims that Tin binding is less dependent on sequence as binding complexity increases. For example, no statistical test was provided for the motif density comparison in subsection “TF binding site divergence is tolerated in the context of collective TF binding”. Please provide the reader with more information as to how to compare the numbers of red/green dots in Figure 4; there seems to be different numbers of data points in the two plots and fractions are not reported/compared.*

We have now added the result of the binomial test assessing if the observed motif density at high-bound Tin enhancers equals that at singletons (subsection “TF binding site divergence is tolerated in the context of collective TF binding”). We also noted that the reported motif densities (7.6 and 10.6, for high-bound and singleton enhancer, respectively) were slightly wrong and were corrected to 7.96 and 10.41. We apologize for this error (but are thankful to have found it). Importantly, with these corrected numbers, the difference in motifs densities is highly significant – p-value < 5e-7, binomial test.

We have now added the numbers of red and green peaks in the Figure 4 legend, and more details on the analysis presented in this new Figure 4.

Briefly, the plot shows 150 singleton and 488 high-bound peaks (as delta ChIP scores must be computed per peak (as measured) and not per enhancer):

· “High dChIP & Low dSTAP” population (green dots): 2 and 21 peaks selected for singleton and high-bound peaks, respectively (p=0.06 in one sided Fisher exact);

· “Low dChIP & High dSTAP” population (red dots): 0 and 28 peaks selected for singleton and high-bound peaks, respectively (p=0.0005 in one sided Fisher exact).

7) It would be interesting to know if the SVM from Zinzen 2009 already accounts for the fact that differential TF occupancy can lead to the same expression output in the manner observed here. For example, if the SVM were given the binding profile of the D. vir CRM 10323, would it predict SM expression? Or given the D. vir CRM 2469 binding profile, would it predict VM expression? This would indicate whether this divergence in TF binding leading to the same outcomes is already observed within D. melanogaster or whether new and unique regulatory logic is being employed in D. virilis, which may indicate the involvement of other TFs.

We concur with the reviewer and we have started an analysis along this line as part of another study with a new student in the group. Briefly, The SVM is able to recapitulate fully or partially the correct expression given *D. virilis* TF binding profiles in approximately 66% of the cases. Although high, this analysis shows that the expression of a substantial number of CRMs (34%) is not faithfully predicted. This may be biological due to divergence in activity, or technical, due to the fact that the SVM training is based on *D. melanogaster* enhancers and the test is based on *D. virilis* binding data. Our current study shows that for some enhancer’s their output can be conserved despite a change in TF binding. Given this, we anticipate that training using *D. virilis* TF occupancy and *D. melanogaster* activity will impact the accuracy of the SVMs predictions, however we need to explore this further. The *D. melanogaster* SVM is also not fully applicable to the *D. virilis* ChIP data, as we have one less time-point for Tinman ChIP in *D. virilis* (the first *D. mel* time point) and thus a new SVM would need to be recomputed. If we were to initiate this today, we would explore different machine learning approaches, which is by itself a major new study.